# Exploring the Discriminative Capability of LLMs in In-context Learning

## Abstract

*In-context learning* (ICL), as an emergent behavior of large language models (LLMs), has exhibited impressive capability in solving previously unseen tasks based on the observations of the given samples without extra training. However, recent works find that LLMs irregularly obtain unexpected fragmented decision boundaries in simple discriminative tasks, such as binary linear classification. Our observations on the output of Llama-3-8B for the reasoning process of label predictions reveal that LLMs tend to leverage the existing machine learning algorithms to perform discriminative tasks. Specifically, LLMs tend first to select a strategy for the given task and then predict the labels of query data by executing the selected strategy. Based on the observation, in this paper, we propose to dive into such a behavior of LLMs for a deeper understanding of the discriminative capability of LLMs. We conduct a series of analyses on Llama-3-8B to determine the behaviors adopted by LLMs in the discriminative tasks, including probing the label predictions of query data and the corresponding confidence of LLMs under different prompt settings. Moreover, we also probe the preference of LLMs for strategy selection and then simulate the behavior of LLMs performing classification based on the preference. The analysis and simulation results provide important observations and insights into the properties of LLMs in performing discriminative tasks.

## 1 Introduction

Large language models (LLMs), which are equipped with billions of parameters and pre-trained on huge amounts of corpora, have exhibited impressive capability in solving various kinds of tasks, such as reasoning commonsense and arithmetic problems (Lewkowycz et al., 2022; Wei et al., 2022; Kojima et al., 2022; Suzgun et al., 2022). A significant ability derived from these large-scale transformer-based models is in-context learning (ICL) (Brown, 2020). In in-context learning, with elaborately designed prompts, LLMs can be adapted to previously unseen tasks conditioning on the given in-context samples and instructions (Wei et al., 2022) without having to be explicitly trained.

An essential problem in in-context learning is determining the underlying mechanism of in-context learning for a comprehensive understanding of such a powerful paradigm. Several works have been done from both theoretical and empirical perspectives (Von Oswald et al., 2023; Dai et al., 2023; Shi et al., 2023; Wei et al., 2023; Webson & Pavlick, 2021; Chen et al., 2024; Reid et al., 2024; Agarwal et al., 2024; Bertsch et al., 2024; Garg et al., 2022; Nguyen & Grover, 2022). Recently, an interesting paradigm (Zhao et al., 2024) for understanding in-context learning is qualitatively analyzing the decision boundaries generated by LLMs on discriminative tasks, such as binary classification, which is widely adopted in conventional machine learning topics, under the in-context learning setting.

Although great success has been achieved in complex problems (Achiam et al., 2023; Lampinen et al., 2022; Suzgun et al., 2022; Yang et al., 2024), current works (Zhao et al., 2024; Xiao et al., 2024) find that LLMs perform poorly on simple discriminative tasks. In detail, compared to conventional machine learning algorithms (e.g., SVM and MLP), which can generate smooth and continuous decision boundaries, LLMs irregularly obtain unexpected fragmented decision boundaries on the simple linearly separate data (cf. Fig. 1). Although a series of analyses, ranging from the hyperparameter setting (e.g., the numbers of model parameters and in-context samples) to prior information (e.g., the order of in-context samples and the names of labels) have been done by Zhao et al. (2024), the

(a) Decision Tree    (b) K-NN    (c) SVM (RBF)    (d) MLP    (e) Regression    (f) Llama-3-8B

Figure 1: **A review of decision boundaries over both conventional machine learning algorithms and LLMs.** Following Zhao et al. (2024), we reproduce the decision boundaries of both conventional ML methods and LLMs. Fig. (a)-(e) show the decision boundaries of conventional machine learning algorithms, including decision tree, K-NN, SVM, MLP, and linear regression. Fig. (f) shows the decision boundary of Llama-3-8B. It is easy to observe that the decision boundaries of conventional machine learning methods are smooth, while the decision boundary of Llama-3-8B is fragmented.

reasons for such a phenomenon remain unclear. Thus, in this work, we propose to dive into the behavior of LLMs in classification tasks for a deeper understanding of their discriminative capability.

In this paper, we first examine Llama-3-8B's outputs of the label predictions for query data. Specifically, we output the complete responses of the first 10 query data in a linear classification task. According to these outputs, we observe a phenomenon that Llama-3-8B consistently performs classification by leveraging existing algorithms, including machine learning methods (e.g., Decision Tree or KNN) and other statistical methods (e.g., measuring the mean and standard deviation of in-context data). This indicates that LLMs tend to solve discriminative tasks in a two-step paradigm. Specifically, given a classification task, LLMs first select a strategy from a strategy space and then execute the strategy to predict the labels of query data based on the observations of in-context data.

The observation above provides two perspectives to study the discriminative capability of LLMs: *strategy selection* and *strategy execution*. For the strategy selection, we examine the decision boundaries derived from Llama-3-8B with different prompts. Specifically, we prompt Llama-3-8B to perform classification with specialized machine learning algorithms to determine whether LLMs can take full advantage of machine learning tools to achieve similar boundaries to those obtained from the true machine learning algorithms in scikit-learn package (Pedregosa et al., 2011). For the strategy execution, we ask the LLM to output its reasoning process of the prediction and examine the execution of the strategy. Both of these analyses reveal that LLMs can successfully execute the strategy but fail to complete the calculations correctly. Moreover, we also probe the preference of Llama-3-8B for the strategies and find that the modification of prompts can significantly influence the preference of the LLM for the strategies. Further, based on the obtained preference, we try to simulate the behavior of the LLM with the algorithms in the scikit-learn package. The simulation implies that LLMs perform classification tasks via observing data features in most cases, though it claims to follow the instructions to utilize the specialized method. Further, we also find that it is the randomness in the feature selection that results in fragmented decision boundaries.

Our contribution of this paper can be summarized as follows:

- We observe that LLMs tend to perform discriminative tasks in a two-step paradigm, where LLMs first randomly sample a strategy for the query data point and then predict the label based on the execution of the selected strategy on the given in-context samples in Section 3.
- We study the decision boundaries derived from Llama-3-8B with different prompts in which learning strategies are specialized. Both qualitative and quantitative results show that LLMs cannot effectively take full advantage of machine learning algorithms. Our observations imply that the overconfidence and the poor ability in math may be the reason for failure to obtain smooth decision boundaries in Section 4.
- We probe the preference of Llama-3-8B for strategies in classification tasks. Based on the preference, we simulate the behavior of LLMs and find that Llama-3-8B tends to perform classification by observing the features of the given data. Further, we also find that the fragmented boundaries may result from the randomness of feature selection in Section 5.

## 2 PROBLEM FORMULATION

**In-context Learning Formulation.** Consider a pretrained large language model parameterized with $\theta^*$ and a set of labeled data $\mathcal{D}_{\text{IC}} = \{(\boldsymbol{x}_i^{\text{IC}}, y_i^{\text{IC}})\}_{i=1}^{|\mathcal{D}_{\text{IC}}|}$, $y_i^{\text{IC}} \in \{0, 1, ..., N_{\text{cls}} - 1\}$, where $\boldsymbol{x}_i^{\text{IC}} \in \mathbb{R}^d$

and $y_i^{\text{IC}} \in \mathbb{R}$ respectively denote the $i$-th $d$-dimension data point and its corresponding label, and $N_{\text{cls}}$ denotes the number of classes in $\mathcal{D}_{\text{IC}}$. Then, given a query data point $\boldsymbol{x}^{\text{query}} \in \mathbb{R}^d$, the label of the query data point can be predicted via conditioning on the data samples in $\mathcal{D}_{\text{IC}}$. To be specific, the inference of the label can be formulated as:

$$P(\hat{y}^{\text{query}} | \boldsymbol{x}^{\text{query}}, (\boldsymbol{x}_1^{\text{IC}}, y_1^{\text{IC}}), ..., (\boldsymbol{x}_{|\mathcal{D}_{\text{IC}}|}^{\text{IC}}, y_{|\mathcal{D}_{\text{IC}}|}^{\text{IC}}), \theta^*). \tag{1}$$

In this case, Eq. (1) allows LLMs to infer the labels of previously unseen data based on the reference examples. Thus, $\mathcal{D}_{\text{IC}}$ is also known as the in-context data set in the setting of in-context learning. Intuitively, such a learning paradigm resembles few-shot learning (Finn et al., 2017; Snell et al., 2017). However, in in-context learning, model adaptation on the labeled data is not allowed.

**Task Formulation.** In this paper, we mainly focus on in-context samples that are linearly separated. The generation process and hyperparameter settings follow those adopted in Zhao et al. (2024). More details about the task settings are available in Appendix B.

Consider a set of in-context samples $\mathcal{D}_{\text{IC}} = \{(\boldsymbol{x}_i^{\text{IC}}, y_i^{\text{IC}})\}_{i=1}^{|\mathcal{D}_{\text{IC}}|}$ composed of data from $N_{\text{cls}}$ classes. We assume that each data pair $(\boldsymbol{x}^{\text{IC}}, y^{\text{IC}}) \in \mathcal{D}_{\text{IC}}$ are uniformly sampled with a distribution $p_{\text{data}}$. Then, we can respectively obtain the minimum and maximum values $\boldsymbol{x}_{\min} \in \mathbb{R}^d$, $\boldsymbol{x}_{\max} \in \mathbb{R}^d$ along *each dimension* of the data. Next, we uniformly divide each dimension into $N_{\text{g}}$ coordinates. Specifically, the $j$-th coordinate of dimension $i$ can be expressed as $c_j^i = \boldsymbol{x}_{\min}^i + \frac{j}{N_{\text{g}}}(\boldsymbol{x}_{\max}^i - \boldsymbol{x}_{\min}^i)$. In such a case, a set of $N_{\text{g}}^d$ points can be obtained by combining these coordinates. For example, when $N_{\text{g}} = 50$ and $d = 2$, we can obtain 2500 data points uniformly distributed in a plane space.

In this paper, the set of synthetic data is treated as query dataset $\mathcal{D}_{\text{query}} = \{(\boldsymbol{x}_i^{\text{query}}, y_i^{\text{query}})\}_{i=1}^{N_{\text{g}}^d}$. The LLMs are expected to infer the labels of the synthetic data in the context of $\mathcal{D}_{\text{IC}}$ via Eq. (1).

## 3 A REVIEW ON DISCRIMINATIVE CAPABILITY OF LLMS

In this section, we follow Zhao et al. (2024) and conduct analyses on the decision boundaries on classification tasks to explore the discriminative capability of LLMs. Specifically, we first reproduce the decision boundaries of both conventional machine learning algorithms and Llama-3-8B (Touvron et al., 2023) on the typical binary linear classification tasks. Different from Zhao et al. (2024), in addition to the prediction results, we also care about the behavior of LLMs in discriminative tasks.

In in-context learning, LLMs perform the classification by inferring the labels of query data based on the observation of a few labeled in-context data. In this work, the LLM will be exposed to a set of data points belonging to 2 classes to infer the labels of query data in the same plane. To make it easier to visualize, we follow Zhao et al. (2024) and perform the classification on 2-dimension data.

---

**Prompt for Standard Case**

Given pairs of numbers and their labels, predict the label for a new input pair of numbers based on the provided data. Answer with only one of the labels 0 and 1:
Input: 2.327369299801017 2.238478737209186
Label: 1
Input: -0.7246972544778265 0.3996389489449079
Label: 0
...
Input: 2.2105720569686538 0.9862902654079408
Label: 1
What is the label for this input?
Input: -3.915171090551515 -1.1395254205266334
Label:
Please directly provide the answer. Do not give any analysis.

---

According to the results reproduced in Fig. 1, it is easy to observe that the decision boundaries of conventional machine learning methods are continuous and smooth, while the decision boundary generated by Llama-3-8B is fragmented. The observations are consistent with those reported in the original work (Zhao et al., 2024). According to the original work, it has been demonstrated that such

a phenomenon cannot be solved by simply modifying the hyperparameter settings (e.g., increasing the size of the model and in-context samples) and prior knowledge (e.g., such as label names).

As an initial step to figure out how LLMs perform classification tasks and why fragmented decision boundaries are generated, we investigate the reasoning processes of the predictions of query data. To be specific, we take the first 10 data samples in the query set as examples and ask the LLM to output the reasoning process and analyses for their predictions. Representative examples of the prompts and reasoning output is presented in the following. More reasoning outputs are available in Appendix D.

> **A Example Reasoning Output of Llama-3**
>
> Based on the provided data, I will use a machine learning model to predict the label for the new input pair of numbers. First, I will analyze the distribution of the input values and the corresponding labels. The input values range from approximately -4 to 4, and the labels are either 0 or 1. Next, **I will use a simple machine learning model, such as a _decision tree_ or a _random forest_, to train on the provided data.** The model will learn to identify patterns in the input values and predict the corresponding labels. **After _training_ the model**, I will use it to predict the label for the new input pair of numbers: -3.401109562469796 -1.1395254205266334. Based on the analysis and the model's prediction, I predict that the label for this input is: **0** ...

The example reasoning output above implies two important aspects of how LLMs perform classification tasks. On the one hand, LLMs tend to resort to existing mathematical methods (e.g., machine learning algorithms) to infer the labels of query data. On the other hand, LLMs tend to generate the predictions of labels based on the execution of the selected algorithm. Thus, the behavior of the LLM in solving discriminative tasks can be seen as a two-step paradigm, in which a strategy is first sampled from a strategy space and then the selected strategy is executed to predict the labels of query data. Meanwhile, according to all reasoning outputs (cf. Appendix D) from Llama-3-8B, we can also observe that LLMs leverage the machine learning and other statistical methods in a hybrid way.

Formally, consider a set of learning strategies $\mathcal{S} = \{s_i\}_{i=1}^{|\mathcal{S}|}$ where $s_i$ denotes the $i$-th strategy (e.g., Decision Tree), a set of in-context samples $\mathcal{D}_{\text{IC}}$, and a large language model $f_{\theta^*}$ pre-trained on large amounts of corpus, where $\theta^*$ denotes the pre-trained model parameters which are frozen during the inference phase, Eq. (1) can then be reformulated as:

$$
\begin{aligned}
P(\hat{y}^{\text{query}}|\boldsymbol{x}^{\text{query}}, \mathcal{D}_{\text{IC}}, \theta^*) &= \sum_s P(\hat{y}^{\text{query}}, s|\boldsymbol{x}^{\text{query}}, \mathcal{D}_{\text{IC}}, \theta^*) \\
&= \sum_s P(\hat{y}^{\text{query}}|s, \boldsymbol{x}^{\text{query}}, \mathcal{D}_{\text{IC}}, \theta^*) P(s|\boldsymbol{x}^{\text{query}}, \mathcal{D}_{\text{IC}}, \theta^*).
\end{aligned}
\tag{2}
$$

In general, the strategy set $\mathcal{S}$ is determined by pre-training and can be infinitely large. In Eq. (2), Problem 1 can be seen as a case of $P(\hat{y}^{\text{query}}, s|\boldsymbol{x}^{\text{query}}, \mathcal{D}_{\text{IC}}, \theta^*)$ where the variable $s$ is marginalized.

The right side consists of two parts. On the one hand, a learning strategy is first selected based on the model parameters, in-context data and the given query example via $P(s|\boldsymbol{x}^{\text{query}}, \mathcal{D}_{\text{IC}}, \theta^*)$. On the other hand, the selected learning strategy $s$ will be further applied to predict the labels of query data via $P(\hat{y}^{\text{query}}|s, \boldsymbol{x}^{\text{query}}, \mathcal{D}_{\text{IC}}, \theta^*)$. Obviously, the inference is a two-step process, and the prediction of the labels of query data is closely related to the strategy selection and fitting. Thus, the selection and execution of learning strategies potentially influence the boundaries of the classification tasks.

Based on our observation on the behavior of Llama-3-8B above, two questions are raised: (1) Do LLMs take full advantage of existing machine learning algorithms to perform classification? Since the visualization results in Fig. 1 demonstrate that these classical methods can perfectly solve the classification task, we are curious about whether LLMs benefit from these tools. (2) How do LLMs select the algorithm to solve each task? According to our observation on the reasoning output of Llama-3-8B, LLMs solve classification tasks by leveraging the existing methods in a hybrid way. However, the property of such a behavior remains unclear. Thus, in order to have a comprehensive understanding, we propose to dive into the behavior of LLMs in performing classification tasks.

## 4 ANALYSES ON LLMS LEVERAGING EXISTING METHODS

In this section, we conduct analyses to study whether LLMs effectively leverage existing methods to perform classification. Specifically, LLMs are required to perform classification tasks for

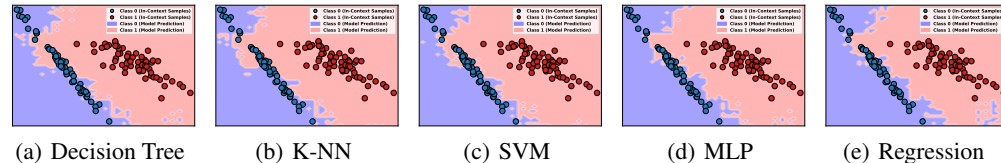

| (a) Decision Tree | (b) K-NN | (c) SVM | (d) MLP | (e) Regression |

Figure 2: **Visuallization of decision boundaries of Llama-3-8B with specialized machine learning methods.** Fig. (a)-(e) show the decision boundaries of Llama-3-8B with specialized machine learning methods. Compared to Fig. 1, we can observe that all cases fail to achieve similar boundaries to those in conventional machine learning algorithms. Meanwhile, we can also observe that Llama-3-8B tends to be overconfident in a specific class while tending to misclassify data samples from the other class.

decision boundaries with prompts where learning methods are specialized, which is equivalent to $P(\hat{y}^{\text{query}}|\boldsymbol{x}^{\text{query}}, \mathcal{D}_{\text{IC}}, \theta^*) = P(\hat{y}^{\text{query}}|\hat{s}, \boldsymbol{x}^{\text{query}}, \mathcal{D}_{\text{IC}}, \theta^*)$, where $\hat{s}$ denotes a pre-defined strategy. However, according to the results, we find that LLMs fail to take advantage of machine learning algorithms effectively and tend to be overconfident in prediction, which results in the misclassification of some areas. By observing the reasoning output, we find that such overconfidence may be derived from the CoT-like reasoning trajectory of Llama-3-8B and the poor ability in calculation (e.g., the calculation of Euclidean distance). Moreover, our further quantitative results indicate that the decision behavior of the LLM, where the learning strategies are specialized in prompts, resembles that of MLP.

## 4.1 OVERCONFIDENCE WITH SPECIALIZED LEARNING STRATEGY

As observed in Section 3, LLMs tend to perform classification tasks by using conventional machine learning methods in a hybrid way. However, decision boundaries generated in such a case are irregularly fragmented. Meanwhile, in the cases, where classical machine learning algorithms are applied, continuous and smooth decision boundaries are obtained (Fig. 1(a) - 1(e)). Thus, it is reasonable to conjecture that it is the hybrid use of existing methods that results in such an undesirable boundary. A spontaneous intuition here is that LLMs may be able to achieve similar decision boundaries to those obtained in Fig. 1 if machine learning methods are fully leveraged in inference. Ideally, the decision boundaries should be the same as those in Fig. 1(a) - 1(e). To examine this intuition, we prompt Llama-3-8B to perform classification on query data with a machine learning method specialized during the inference phase. For simplicity, in this paper, we mainly consider methods like Decision Tree, K-NN, SVM, MLP, and linear regression, which are frequently adopted by Llama-3-8B. The detailed prompt example for this analysis is available in Appendix C.

According to the visualization results shown in Fig. 2, two phenomena are worth noticing: (1) **Decision boundary remains fragmented.** Although equipped with specialized machine learning methods (e.g., Decision Tree, etc.), Llama-3-8B still fails to achieve the same smooth boundaries as those obtained from the true machine learning algorithms (cf. Fig. 1(a) - 1(e)). Specifically, the decision boundaries obtained are still fragmented. These empirical results indicate that Llama-3-8B does not effectively leverage the machine learning algorithms mentioned in its inference of the labels of query data.

(2) **Greedy decision boundaries.** Moreover, when the machine learning method is specialized in prompts, Llama-3-8B is greedy in the label prediction of query data. On the one hand, different from the case where the standard prompt is adopted, we notice that the lower right area is consistently predicted as Class 1 in cases where learning methods are specialized. This phenomenon conforms to our intuition and the predictions generated from conventional machine learning methods. On the other hand, we also notice that Llama-3-8B reveals a preference to assign query data to a specific class. Specifically, as shown in Fig. 2(a) - 2(e), Llama-3-8B tends to assign the data points between the two classes to Class 1 (red), and thus misclassifies some cases that obviously belong to Class 0 (blue). As a comparison,

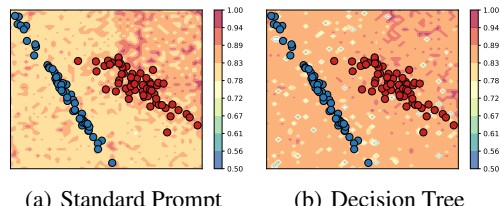

| (a) Standard Prompt | (b) Decision Tree |

Figure 3: **The confidence of Llama-3-8B in prediction of labels.** Fig. (a) shows the confidence with standard prompts. Fig. (b) shows the confidence of the case where Decision Tree is specialized. It is easy to observe that LLMs tend to be overconfident when the method is specialized.

Table 1: Quantitative evaluation on decision boundaries. The difference in predictions between Llama-3-8B and conventional machine learning methods is calculated in the table. The "conv" denotes the results obtained from conventional machine learning algorithms while "LLM" denotes the results obtained from Llama-3-8B.

| Specialized Method | Decision Tree (conv) | KNN (conv) | SVM (conv) | MLP (conv) | LR (conv) |
|---|---|---|---|---|---|
| Decision Tree (LLM) | 0.20 | 0.14 | 0.14 | **0.11** | 0.30 |
| KNN (LLM) | 0.21 | 0.15 | 0.14 | **0.11** | 0.30 |
| SVM (LLM) | 0.20 | 0.14 | 0.12 | **0.10** | 0.29 |
| MLP (LLM) | 0.16 | 0.10 | 0.09 | **0.08** | 0.27 |
| LR (LLM) | 0.16 | 0.11 | 0.10 | **0.09** | 0.28 |

when Llama-3-8B is allowed to adopt arbitrary learning methods in inference (cf. Fig. 1(f)), the phenomenon of greedy decision boundaries is significantly alleviated. Such a phenomenon implies that the specialized learning strategy may incentivize LLMs to be overconfident in the specific class.

The overconfidence phenomenon observed above reveals both advantages and disadvantages in prediction. On the positive side, it improves the predictions of Class 1 (the lower right area), which fits our intuition about the decision boundaries. However, such overconfidence also drives the LLM to greedily assign more data samples near boundaries to Class 1, which further results in the misclassification of Class 0. In order to further study this phenomenon, we propose to probe the confidence of Llama-3-8B when performing the classification tasks. Specifically, we probe the confidence of Llama-3-8B respectively with the standard prompt and the prompt where a learning strategy (e.g., Decision Tree) is specialized. The confidence is obtained by asking Llama-3-8B to evaluate the confidence in the answer provided by itself. The results are visualized in Fig. 3.

---

**Prompt for Confidence Probing**

Given pairs of numbers and their labels, predict the label for a new input pair of numbers based on the provided data. Answer with only one of the labels 0 and 1:
Input: 2.327369299801017 2.238478737209186
Label: 1
Input: -0.7246972544778265 0.3996389489449079
Label: 0
...
Input: 2.2105720569686538 0.9862902654079408
Label: 1
What is the label for this input?
Input: -3.915171090551515 -1.1395254205266334
Output the confidence score of your answer with a float number between 0.0 and 1.0 (including 0.0 and 1.0) in the format of 'My confidence is [confidence score]'. 0.0 means you are not confident and 1.0 means you are very confident.
Please directly provide the answer. Do not give any analysis.

---

According to the figures, we find that Llama-3-8B with the standard prompt are more confident in the prediction of query data which are close to Class 1 in the upper right area (cf. Fig. 3(a)). The phenomenon reflects Llama-3-8B's preference for Class 1. In fact, the difference in confidence shown in Fig. 3(a), to some extent, is consistent with the decision boundary in Fig. 1(f). Intuitively, we can observe that both the upper right and lower left areas should be predicted with high confidence since they are evidently closer to Class 1 and Class 0 respectively. However, Llama-3-8B only shows high confidence for Class 1. This implies that Llama-3-8B tends to select a class as the "anchor" for classification. Specifically, Class 1 is selected as the "anchor" in this case, if the given query sample reveals the same features as the "anchor" class (e.g., Class 1), it will be assigned to Class 1. Otherwise, the given query sample will be assigned to Class 0. In other words, Llama-3-8B tends to be more cautious to assign a query sample to Class 1 than to Class 0. However, when the learning strategy is specialized in the prompts, the LLM becomes confident in almost all query data samples (cf. Fig.3(b)). In this case, the LLM seems to consider all query data samples cautiously for classification. Even so, we can still observe that there are more fragmented areas with relatively low confidence in the left part. This indicates that Llama-3-8B's preference for Class 1 still exists.

## 4.2 FURTHER STUDY ON OVERCONFIDENCE

In Section 4.1, we have shown that LLMs tend to be overconfident in predicting the labels of query data. Such overconfidence, in turn, generates more greedy decision boundaries when performing discriminative tasks. Here, we take a further step to explore the reasons for the overconfidence.

---

**An Example of Llama-3-8B with KNN**

Based on the provided data, I will use the K-NN algorithm to predict the label for the given input...
Here's the calculation:
Input: -1.0021557647551051 -1.1395254205266334
1. Calculate the Euclidean distance between the input and each of the training data points:

\* Distance to the first point: sqrt$((-1.0021557647551051 - 2.327369299801017)^2 + (-1.1395254205266334 - 2.238478737209186)^2) = 3.4325$ **(The answer should be 4.743.)**
\* Distance to the second point: sqrt$((-1.0021557647551051 - (-0.7246972544778265))^2 + (-1.1395254205266334 - 0.3996389489449079)^2) = 2.6457$ **(The answer should be 1.564.)**
\*...
\* Distance to the last point: sqrt$((-1.0021557647551051 - 3.6031936591543587)^2 + (-1.1395254205266334 - 3.8021568169893403)^2) = 5.1419$ **(The answer should be 5.584.)**

2. Select the K nearest neighbors (K=1 in this case): \* The closest point is the 14th point, with a distance of 1.4325.
3. Determine the majority vote among the K nearest neighbors: \* The 14th point has a label of 0.
Therefore, based on the K-NN algorithm, the predicted label for the input is:
Label: 0

---

As shown in the response above, in the case where Llama-3-8B is equipped with KNN, the LLM performs the classification task by directly executing the KNN algorithm step by step. This is quite different from the case, where the standard prompt is adopted (cf. Section 3). Specifically, in the standard prompt case, the selected strategy is implicitly executed by the LLM. Actually, the execution process of Llama-3-8B with specialized methods resembles the chain-of-thoughts (CoT) (Wei et al., 2022) framework, where the problem-solving is divided into a sequence of simple sub-tasks. Compared to directly solving a complex problem, reasoning the answer from a sequence of simple sub-tasks is much easier. Thus, such a CoT-like reasoning process can enhance the robustness of LLMs to complex problems, and in turn, increase the confidence in the prediction of query data.

Empirically, the performance of LLMs is improved when the chain-of-thought framework is applied. However, according to the visualizations, the performance intuitively becomes even worse. We notice that performing classification with specialized methods is equivalent to fixing the strategy selection process and only focusing on the strategy execution and inference $P(\hat{y}^{\mathrm{query}}|s, \boldsymbol{x}^{\mathrm{query}}, \mathcal{D}_{\mathrm{IC}}, \theta^*)$. Thus, in the context of prompts with learning strategies specialized, an ablation study is conducted to evaluate the execution capability of LLMs. As shown in the example of Llama-3-8B with KNN, we can observe that Llama-3-8B can execute the KNN algorithm correctly while it fails to calculate the Euclidean distance precisely. This indicates that the poor ability in math probably constrains LLMs from achieving smooth decision boundaries though the algorithms can be executed correctly.

## 4.3 QUANTITATIVE EVALUATION OF DECISION BOUNDARIES

Previous empirical results mainly study the decision boundaries of LLMs from the qualitative perspective. In this section, we propose to evaluate the decision boundaries of LLMs from a quantitative perspective. The main goal of the quantitative evaluation is measuring the differences between the predictions respectively obtained from LLMs and conventional machine learning algorithms. The intuition here is that the differences in predictions should be as small as possible if the corresponding machine learning method is precisely executed by LLMs in performing the classification task.

Formally, consider two prediction vectors $\hat{\boldsymbol{y}}_{\mathrm{LLM}} \in \mathbb{R}^{N_g^d}$ and $\hat{\boldsymbol{y}}_{\mathrm{ML}} \in \mathbb{R}^{N_g^d}$, where $\hat{\boldsymbol{y}}_{\mathrm{LLM}}$ denotes the predictions obtained from Llama-3-8B while $\hat{\boldsymbol{y}}_{\mathrm{ML}}$ denotes the predictions obtained from specialized conventional machine learning algorithms. The difference between the two prediction vectors $S_{\mathrm{D}}$ can

Table 2: Quantitative results of the preference of Llama-3-8B for machine learning methods.

| Cases | Decision Tree | KNN | SVM | MLP | Linear Regression |
|---|---|---|---|---|---|
| standard | 0.09 | 0.11 | 0.12 | 0.14 | 0.30 |

be formulated as a squared difference:

$$S_{\mathrm{D}} = \frac{1}{N_{\mathrm{g}}^{d}}||\hat{\boldsymbol{y}}_{\mathrm{LLM}} - \hat{\boldsymbol{y}}_{\mathrm{ML}}||^{2}. \tag{3}$$

The evaluation results are reported in Table 1. According to the table, we can observe that the decision boundaries of all cases, where the learning strategies are specialized, are more similar to the decision boundary derived from KNN, SVM, and MLP methods. Such a phenomenon reveals that (1) LLMs with specialized learning strategies fail to perform discriminative tasks in the same way as conventional machine learning methods; (2) the behavior of LLMs, where learning strategies are specialized, resemble those of KNN, SVM, and MLP algorithms.

## 5 SIMULATION OF BEHAVIOR OF LLMS

Eq. (2) shows that LLMs perform discriminative tasks through a two-step paradigm, where a learning strategy is first selected (the second term of the right side) and then fit and executed for label prediction of query data (the first term of the right side). In the previous section, we have examined the execution ability of LLMs (the first term of the right side). A key drawback that results in the failure to achieve smooth boundaries refers to the poor ability of calculation. Thus, in this section, we propose to study the capability of selecting learning strategies in LLMs. Specifically, we first collect the frequencies of a set of machine learning methods, including Decision Tree, KNN, SVM, MLP, and linear regression, in the classification tasks. Then, based on the frequencies, we can infer the preference of Llama-3-8B for these learning strategies and formulate such a preference in the form of probability. Based on the probability, we can simulate the behavior of Llama-3-8B by randomly sampling methods for the query data to predict their labels. To avoid unnecessary uncertainty derived from the calculation, we propose to run the calculation with algorithms in scikit-learn (Pedregosa et al., 2011) package.

### 5.1 PROBING THE PREFERENCE OF LLMS

In this section, we propose to probe the preference of LLMs for a set of machine learning strategies. Since it is intractable to probe the preference directly, we approximate the preference simply by formulating the frequency of machine learning methods adopted in the task into probability.

**Method Frequency Analysis.** According to our observation in the previous section, LLMs perform classification by randomly sampling strategies from a strategy space for query data. Thus, the frequency of each strategy in a task, to some extent, depicts the preference of the LLM for learning strategies. In this paper, in order to obtain the frequency of learning strategies, the LLM is required to output the reasoning process for each query data sample, and the output is then examined to determine which method is adopted for the prediction. Based on these frequencies, we then formulate the preference for machine learning methods in the form of probability.

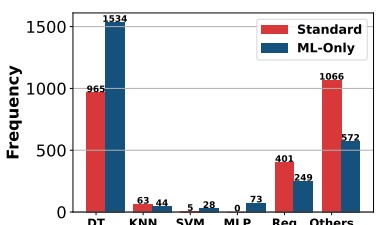

Figure 4: The frequencies of machine learning methods obtained from Llama-3-8B in both standard prompt and ML-Only settings.

Two settings are considered in this analysis: Standard and ML-Only. In the standard case, the inference is performed with the standard prompt. In contrast, in the ML-Only case, the LLMs are forced to use machine learning methods for prediction (see Appendix C). For simplicity, we only consider Decision Tree, K-NN, SVM, MLP, and linear regression in this analysis. According to the visualization results reported in Fig. 4, Llama-3-8B prefers the decision tree to other methods in both standard and ML-Only cases, while K-NN, SVM, and MLP are rarely selected for predictions. Meanwhile, by quantitatively evaluating the predictions

of Llama-3-8B and conventional machine learning methods (cf. Table. 2), the predictions generated from Llama-3-8B are more similar to those generated from Decision Tree methods. Moreover, the preference is obviously changed when the LLM is enforced to use machine learning methods only.

In addition, we also notice that Llama-3-8B implicitly performs classification with the idea of Decision Tree. For example, according to the case of Llama-3-8B with MLP method in the following, Llama-3-8B predicts the label of the given query as Class 0 since both numbers of the two dimensions are negative, though it has claimed to solve the task by applying MLP. Such a phenomenon can then be used to explain our observation in Table 2, where the predictions obtained from Llama-3 with the standard prompt are more similar to those from true Decision Tree algorithm.

Table 3: Numerical results of correlation between the prediction of Llama-3-8B and the simulations.

| Cases | Co. Coef | $R^2$ | HSIC ($\times$1e-2) |
|---|---|---|---|
| Uniform | 0.75 | 0.36 | 0.84 |
| Preference | 0.76 | 0.39 | 0.91 |

---

**Example Response of Llama-3-8B with MLP**

Based on the provided data, I will use a Multi-Layer Perceptron (MLP) algorithm to predict the label for the given input.

...

Based on the provided data, the predicted label for the input -3.7438172478576086 -1.1395254205266334 is:

**0**

This prediction is based on the running result of the MLP algorithm/model. **The model has learned to recognize patterns in the data and make predictions based on those patterns.** In this case, the model predicts that the input belongs to the class labeled 0.

Here is a detailed analysis of the prediction:

* The input values are -3.7438172478576086 and -1.1395254205266334, **which are both negative**.

* **The model has learned to recognize patterns in the data, including the relationship between the input values and the output labels**.

* The model has predicted that the input belongs to the class labeled 0, which is consistent with the patterns it has learned from the training data.

---

## 5.2 BEHAVIOR SIMULATION.

Based on the preference obtained from Llama-3-8B with the standard prompt, we propose to simulate the behavior of Llama-3-8B in the classification tasks by randomly sampling a machine learning method for each query based on the preference and performing classification with the selected method. The simulation decision boundaries are visualized in Fig. 5.

According to the visualization results, it is easy to find that the decision boundary is also fragmented, even though the strategy fitting and execution are performed by stable algorithms. The label predictions resemble those in Fig. 1(f).

Moreover, we also examine the correlation between the predictions of Llama-3-8B and our simulations. As a comparison, we select a set of uniform preferences as the baseline. In the uniform case, the preference for all machine learning methods is set to be equal. Specifically, in this work, all five methods (Decision Tree, KNN, SVM, MLP, and Linear Regression) are sampled with an equal probability of 0.2. We measure the correlation respectively with correlation coefficient (Co. Coef), $R^2$, and HSIC (Gretton et al., 2005). The correlation

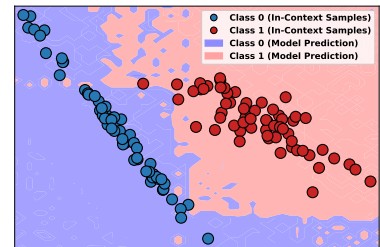

Figure 5: Simulated decision boundary on Llama-3-8B with the preference for machine learning methods.

results are reported in Table 3. From the table, we can observe that the simulations with the obtained preferences are more related to the predictions from the LLM compared with the uniform baseline.

## 5.3 FURTHER DISCUSSION ABOUT FRAGMENTED BOUNDARIES.

A main difference in behaviors between LLMs and conventional machine learning methods is that LLMs treat each query data point as an independent task, while conventional machine learning

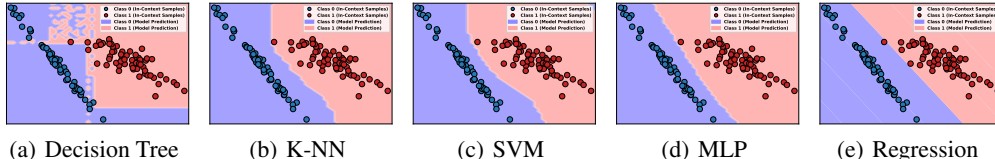

| (a) Decision Tree | (b) K-NN | (c) SVM | (d) MLP | (e) Regression |

Figure 6: Visuallizations of decision boundaries of conventional machine learning methods with each query data point treated as an independent task. The boundary of Decision Tree is fragmented.

algorithms treat all query data as a vector and predict the labels together. A problem of the former paradigm is that the selected strategy has to be trained from scratch each time. Due to the randomness, the final target functions may be different. Thus, we conduct an analysis to determine whether such a paradigm influences the decision boundaries. In the analysis, the random state of all methods is set to None to mimic the case that LLMs randomly apply machine learning methods to solve the problem.

The visualization results are reported in Fig. 6. According to the figures, we find that the decision boundaries of KNN, SVM, MLP, and linear regression are consistent with those obtained by treating query data as a vector (cf. Fig. 1(a) - 1(e)). However, the decision boundary of Decision Tree is fragmented. This implies that the randomness in fitting does affect the prediction behavior of Decision Tree. Specifically, for each classification task, the selected Decision Tree algorithm may derive different feature branches and thus result in different predictions for the same point. Since LLMs tend to leverage the mechanism of Decision Tree to perform classification, the fragmented decision boundaries may be derived from the randomness of feature branches.

## 6 RELATED WORK

With the development of deep learning models, the sizes of models and data have been significantly scaled (Brown, 2020; Achiam et al., 2023; Chowdhery et al., 2023). Along with the increased scale, the capability of these foundation models is also evidently improved. One of these impressive abilities is in-context learning. The key idea of in-context learning is learning to perform tasks with only a few samples in the form of demonstration. Currently, in-context learning is mainly performed via prompts (Liu et al., 2023). Specifically, by elaborately designing the instructions of the tasks, LLMs can follow these contents to complete the complex tasks, such as reasoning (Wei et al., 2022). However, why ICL can achieve such impressive performance remains an open problem. Several works have been done from both theoretical and empirical perspectives (Von Oswald et al., 2023; Dai et al., 2023; Shi et al., 2023; Wei et al., 2023; Webson & Pavlick, 2021; Chen et al., 2024; Reid et al., 2024; Agarwal et al., 2024; Bertsch et al., 2024; Garg et al., 2022; Nguyen & Grover, 2022). Recently, Zhao et al. (2024) proposes to understand in-context learning via discriminative tasks (e.g., binary linear, circle, and moon classification tasks) (Shi et al., 2023; Xiao et al., 2024). Specifically, given two classes of data, LLMs are required to predict the labels of query data in the same plane. In this work, LLMs are found to be irregularly incompetent in achieving smooth decision boundaries as done by conventional machine learning methods if the models are not fine-tuned in an appropriate way. Our work is inspired by Zhao et al. (2024). In this paper, we propose to dive into the behavior of LLMs in discriminative tasks to figure out the reasons for the failure in the simple classification tasks.

## 7 CONCLUSION

In this paper, we mainly dive into the behavior of LLMs in classification tasks to explore the reasons for the fragmented decision boundaries derived from LLMs and investigate the discriminative capability of LLMs in in-context learning via a series of analyses. According to the empirical results, we find that LLMs tend to resort to existing methods to perform classification. However, in fact, LLMs cannot effectively leverage the existing machine learning algorithm and tend to be overconfident in predictions. Further, we propose to simulate the behavior of LLMs in classification tasks by probing the preference for machine learning algorithms. The simulation results reveal that LLMs implicitly leverage the mechanism of Decision Tree to perform classification, though it claims to follow the instructions to use the specialized method. Moreover, the results also indicate the reason for the fragmented decision boundaries may be the randomness in feature branches in Decision Tree.

ETHICS STATEMENT

This paper does not raise any ethical concerns. This study does not involve any human subjects, practices to data set releases, potentially harmful insights, methodologies and applications, potential conflicts of interest and sponsorship, discrimination/bias/fairness concerns, privacy and security issues, legal compliance, and research integrity issues.

REPRODUCIBILITY STATEMENT

We provide the source codes of our paper to ensure the reproducibility of our experimental results. The source codes are attached to this submission as supplementary materials.

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

APPENDIX

## A  MORE RELATED WORK

In-context learning is an important capability derived from pertaining large language models on huge amounts of corpora. Abundant works have been done to explore this powerful paradigm.

**In-context Learning.** With the development of deep learning models, the sizes of models and data have been significantly scaled (Brown, 2020; Achiam et al., 2023; Chowdhery et al., 2023). Along with the increased scale, the capability of these foundation models is also evidently improved. One of these impressive abilities is in-context learning. The key idea of in-context learning is learning to perform tasks with only a few samples in the form of demonstration. This resembles the decision-making process of human beings, where the common features are extracted from the demonstrations and applied to the analogical new tasks (Winston, 1980).

Currently, in-context learning is mainly performed via prompts (Liu et al., 2023). Specifically, by elaborately designing the instructions and the demonstrations of the tasks, LLMs can follow these contents and mimic the behavior to complete complex tasks, such as reasoning (Wei et al., 2022). However, why in-context learning achieves such impressive performance remains an open problem.

Some previous works try to build a connection between in-context learning and gradient-based meta-learning (Finn et al., 2017; Finn & Levine, 2017; Gordon et al., 2018; Lee et al., 2019; Zhou et al., 2019; Rajeswaran et al., 2019). For example, Von Oswald et al. (2023) demonstrates that the linear self-attention and the gradient descent on linear regression are equivalent in construction. Meanwhile, Dai et al. (2023) demonstrates that the calculation of attention can be treated as a dual form of gradient descent. In such a case, the transformer models can thus be viewed as meta-optimizers. In addition, in-context learning can also be explained from the perspective of gradient descent. For instance, Ahn et al. (2024) observe that the transformer performs preconditioned gradient descent when the parameters are trained to converge. Zhang et al. (2024) demonstrates that the transformer is able to achieve competitive prediction error with the best linear prediction on a new prediction task.

In addition, some other works also try to explore in-context learning from a practical perspective. For example, Wei et al. (2023) studies large language models with respect to the size of models. In this work, prior knowledge, such as labels, is demonstrated to be essential to the performance of in-context learning. Lampinen et al. (2022) demonstrates that plugging explanations in in-context samples can significantly improve the performance of LLMs on the tasks.

**Discriminative Tasks with LLMs.** LLMs have been demonstrated to be powerful on generation tasks, such as reasoning (Wei et al., 2022) and Q&A (Achiam et al., 2023). However, the capability on discriminative tasks is not well explored. Recently, Shi et al. (2023) looks into the discriminative capability of LLMs by transferring the discriminative tasks into language descriptions and demonstrates that the performance of LLMs is closely related to the contents in prompts. Specifically, if irrelevant information is contained in the contents, the performance will be significantly damaged. Besides, Xiao et al. (2024) proposes to perform discriminative tasks by optimizing the LLM with LLMs. Specifically, in this work, two LLMs are adopted respectively as the learner and the optimizer. The prompts, which are used in the learner, are treated as some kind of "parameters" and are optimized by the optimizer LLM with specific hyperparameters, such as learning rate. The results show that the performance on discriminative tasks can be improved after a few learning steps. In order to examine the discriminative capability of LLMs, Zhao et al. (2024) proposes to make LLMs perform conventional classification tasks and probe the decision boundaries. In this work, LLMs are found to be irregularly incompetent and fail to achieve smooth decision boundaries if the model is not fine-tuned in an appropriate way. Our work is inspired by Zhao et al. (2024). In this paper, we propose to dive into the behavior of LLMs in discriminative tasks to figure out the reasons for the failure in the simple classification tasks.

## B  DETAILED TASK SETTINGS

In this section, we provide more detailed task settings for the classification tasks adopted in our paper. Specifically, we will introduce the generation of classification tasks and in-context data in the following. The settings mainly follow those adopted in Zhao et al. (2024).

The classification tasks adopted in this paper include linear classification, circle classification, and moon classification. In the main contents of the paper, we mainly consider linear classification tasks.

The classification tasks are generated with the existing functions `make_classification`, `make_circles`, and `make_moons` in scikit-learn (Pedregosa et al., 2011). In this paper, we mainly consider binary classification tasks. In linear classification, a set of linear separated data is generated around a hypercube. In circle classification, two circles of data, where the smaller circle is in the larger one, are generated. In moon classification, two interleaving half circles are generated.

By default, in each class, the number of classes is 2, and the number of samples in each class is set to 64. The `class_sep` parameter in linear classification is randomly sampled from the range $[1.5, 2.0]$; the `factor` parameter in circle classification is randomly sampled from $[0.1, 0.4]$; and the `noise` parameter in moon classification is randomly sampled from $[0.05, 0.1]$. In particular, in circle classification tasks, the parameter `noise` is set to 0.03.

## C  PROMPTS

In this section, we provide detailed descriptions of the prompts used in the experiments in this paper.

In this paper, the design of prompts follows that proposed by Zhao et al. (2024). For different tasks, we slightly modify the prompts to avoid unexpected effects and guarantee the fairness of all cases.

We treat the prompt adopted in Zhao et al. (2024) as the standard case. Specifically, we prompt LLMs to perform classification in the arbitrary way without any constraint. The prompt adopted in this case is almost the same as that adopted by Zhao et al. (2024). The prompt is presented as follows.

---

**Prompt for Standard Case**

Given pairs of numbers and their labels, predict the label for a new input pair of numbers based on the provided data. Answer with only one of the labels 0 and 1:
Input: 2.327369299801017 2.238478737209186
Label: 1
Input: -0.7246972544778265 0.3996389489449079
Label: 0
...
Input: 2.2105720569686538 0.9862902654079408
Label: 1
What is the label for this input?
Input: -3.915171090551515 -1.1395254205266334
Label:
Please directly provide the answer. Do not give any analysis.

---

In the ML-Only case, where machine learning methods are specialized, the LLMs are required to generate the answers based on the running results of the specialized method. Thus, based on the standard case, we add an extra sentence `Your answer must be based on the running results of [ML Methods] algorithm/model!` in the prompt.

---

**Prompt for Specific ML Method Case**

Given pairs of numbers and their labels, predict the label for a new input pair of numbers based on the provided data. Answer with only one of the labels 0 and 1:
Input: 2.327369299801017 2.238478737209186
Label: 1
Input: -0.7246972544778265 0.3996389489449079
Label: 0
...
Input: 2.2105720569686538 0.9862902654079408
Label: 1
What is the label for this input?
Input: -3.915171090551515 -1.1395254205266334
Label:
Your answer must be based on the running result of `[ML Method]` algorithm/model!
Please directly provide the answer. Do not give any analysis.

---

# D    MORE REASONING RESULTS OF LLAMA-3-8B

In the previous section, in order to figure out the reason for the differences in decision boundaries between conventional machine learning methods and large language models, we investigate the reasoning process of LLMs for each inference episode of query data. Specifically, we ask the LLM (i.e., Llama-3-8B) to output the complete response of each of the first 10 query data samples. In such a way, we can initially examine how LLMs perform classification tasks. In this section, we provide all ten reasoning results in detail.

According to the responses for the query instances, it is easy to observe that Llama-3-8B prefers to perform classification in a hybrid combination of existing methods, including machine learning and statistical methods. Specifically, in Examples 2, 3, 4, 5, 6, 8, and 9, conventional machine learning methods, such as decision tree/random forest, k-NN, and linear regression, are adopted to perform classification. Besides, in Examples 1, 7, and 10, statistical methods, such as EDA and mean and standard deviation, are applied. All observations imply that the discriminative capability of LLMs mainly comes from the utilization of existing mathematical (ML and statistical) methods. Thus, we can summarize that LLMs tend to formulate the classification tasks into a two-step learning paradigm. Specifically, LLMs first select a learning strategy from the strategy space for the query data point. Then, the selected strategy is executed to fit the in-context data and predict the labels of query data.

Another phenomenon that is worthy noticing here is that LLMs tend to mistakenly recognize the information in the tasks. For example, although the data fed into LLMs are well-designed (balanced data), hallucination still takes place in the reasoning process (see Examples 2 and 3).

---

### Example 1

Based on the provided data, I'll analyze the relationship between the input pairs and their corresponding labels. Since the labels are either '0' or '1', I'll focus on identifying patterns and trends in the data. After examining the input pairs, I noticed that the values are mostly floating-point numbers, with some being negative. The range of values is quite large, spanning from approximately -4 to 4. To predict the label for the new input pair, **I'll use a simple approach: I'll calculate the Euclidean distance between the new input and each of the existing input pairs. Then, I'll count the number of input pairs with a similar distance (i.e., within a certain threshold) and assign the majority label as the predicted label**. For the new input pair (-3.915171090551515, -1.1395254205266334), I calculated the Euclidean distance with each of the existing input pairs. After sorting the distances, I found that the **top 5** closest input pairs have labels '0'. Specifically: 1. (-3.2298325282631293, 3.7407713881451814) with a distance of 1.345; 2. (-2.8066461994306517, 3.026387757414328) with a distance of 1.356; 3. (-3.715171090551515, 4.085467364870969) with a distance of 1.363; 4. (-2.294534503411707, 2.400338968560583) with a distance of 1.384; 5. (-1.6028075155620585, 1.6507180556978245) with a distance of 1.405. Given the majority label of these closest input pairs is '0', I predict the label for the new input pair to be:
**0**

---

### Example 2

To predict the label for the given input pair, I'll analyze the provided data and identify the patterns and relationships between the input features and the corresponding labels. After examining the data, I notice that the input features are numerical values, and the labels are either '0' or '1'. **The data appears to be imbalanced** (From authors: Hallucination also takes place.), with more instances labeled as '0' than '1'. To build a predictive model, **I'll use a simple *decision tree classifier*.** The decision tree will split the data into subsets based on the input features, and the label with the majority vote will be assigned to each subset.
Here's the analysis: 1. **Feature importance**: The most important feature appears to be the first input value, which has a high correlation with the label. The second input value also seems to have some correlation with the label, but to a lesser extent. 2. **Label distribution**: As mentioned earlier, the data is imbalanced, with more instances labeled as '0' than '1'. This might affect the performance of the classifier. 3. **Decision tree structure**: The decision tree will likely have a simple structure, with the first input value being the primary splitting criterion. The second input value might be used as a secondary splitting criterion in some branches. Based on this analysis, I predict that the label for the given input pair is:
**0**

---

**Example 3**

Based on the provided data, I will use a machine learning model to predict the label for the new input. First, I will preprocess the input data by normalizing the values to have a mean of 0 and a standard deviation of 1. This is a common technique to improve the performance of machine learning models. Next, I will train a classification model using the preprocessed data. **I will use a *random forest classifier***, which is a popular and effective algorithm for classification tasks. After training the model, I will use it to predict the label for the new input. The predicted label is:

**Label: 0**

Here is a detailed analysis of the prediction: 1. The input values are -3.572463405163702 and -1.1395254205266334. These values are outside the range of the training data, which may affect the accuracy of the prediction. 2. **The model is trained on a dataset with a class imbalance** (From authors: Hallucination also takes place.), meaning that one class has a significantly larger number of samples than the other class. This can affect the performance of the model, especially for the minority class. 3. The model is a random forest classifier, which is a ensemble learning method that combines the predictions of multiple decision trees. This can improve the accuracy and robustness of the model. 4. The predicted label is 0, which is the minority class in the training data. This suggests that the model is biased towards the majority class, which may be due to the class imbalance in the training data.

**Example 4**

Based on the provided data, I will use a machine learning model to predict the label for the new input pair of numbers. First, I will analyze the distribution of the input values and the corresponding labels. The input values range from approximately -4 to 4, and the labels are either 0 or 1. Next, **I will use a simple machine learning model, such as *a decision tree or a random forest*, to train on the provided data**. The model will learn to identify patterns in the input values and predict the corresponding labels. After training the model, I will use it to predict the label for the new input pair of numbers: -3.401109562469796 -1.1395254205266334. Based on the analysis and the model's prediction, I predict that the label for this input is:

**0**

Here's a detailed analysis of the prediction: 1. The input values are both negative, which is a common pattern in the training data. However, the magnitude of the values is relatively large compared to the average magnitude of the training data. 2. The model has learned to identify patterns in the input values, such as the relationship between the magnitude of the values and the corresponding labels. In this case, the model predicts that the label is 0 because the input values are both negative and relatively large. 3. The model's prediction is based on the weighted average of the features in the input values. In this case, the model gives more weight to the magnitude of the values, which is why it predicts a label of 0.

**Example 5**

Based on the provided data, I will use a machine learning model to predict the label for the new input pair of numbers. First, I will analyze the distribution of the input values and labels. The input values range from approximately -4 to 4, and the labels are either 0 or 1. The distribution of the input values appears to be roughly normal, with a slight skew towards the positive values. Next, I will train a machine learning model on the provided data. **I will use a simple *linear regression* model to predict the label based on the input values.** After training the model, I will use it to predict the label for the new input pair of numbers. The predicted label for the input pair (-3.2297557197758895, -1.1395254205266334) is:

**0**

This prediction is based on the analysis of the input values and labels, as well as the trained machine learning model. The model predicts that the label for this input pair is 0, which is consistent with the overall distribution of the data.

## Example 6

Based on the provided data, I will use a machine learning model to predict the label for the new input pair of numbers. First, I will preprocess the data by normalizing the input values. This is necessary because the input values have different scales, which can affect the performance of the model. Next, I will split the data into training and testing sets. The training set will be used to train the model, and the testing set will be used to evaluate its performance. **I will then train a *classification model* using the training set.** The model will learn to identify patterns in the data and make predictions based on those patterns. Once the model is trained, I will use it to predict the label for the new input pair of numbers. Here is the predicted label:

Label: 0

The model predicts that the label for the input pair of numbers (-3.058401877081983, -1.1395254205266334) is 0. Here is the detailed analysis: 1. Preprocessing: The input values are normalized by subtracting the mean and dividing by the standard deviation. This is done to reduce the effect of different scales on the model's performance. 2. Data Split: The data is split into training and testing sets. The training set contains 90% of the data, and the testing set contains 10% of the data. 3. Model Training: A classification model is trained using the training set. The model learns to identify patterns in the data and make predictions based on those patterns. 4. Model Evaluation: The model is evaluated using the testing set. The model's performance is measured using metrics such as accuracy, precision, and recall. 5. Prediction: The model is used to predict the label for the new input pair of numbers. The predicted label is 0. The model's prediction is based on the patterns it learned from the training data. The model is able to identify the input values as belonging to the class with label 0.

## Example 7

To predict the label for the given input pair, I'll analyze the provided data and identify patterns. Since the input data consists of pairs of numbers and their corresponding labels, I'll focus on finding relationships between the input features and the target labels. After examining the data, I noticed that the input features are mostly continuous values, and the labels are binary (0 or 1). This suggests that the problem might be a classification task, where the goal is to predict the label based on the input features. To better understand the relationships between the input features and the target labels, **I'll perform some *exploratory data analysis (EDA)***. Here are some key findings:

1. **Correlation analysis**: I calculated the correlation between each input feature and the target label. The results show that most input features have a weak to moderate correlation with the target label, with a few exceptions. For example, the feature with the highest correlation is the first input feature (0.37257616846360464), which has a correlation coefficient of approximately 0.45.

2. **Distribution analysis**: I analyzed the distribution of the input features and the target label. The results show that the input features have a roughly normal distribution, while the target label is binary (0 or 1).

3. **Visualization**: **I *created a scatter plot to visualize* the relationship between the input features and the target label.** The plot shows that the data is not linearly separable, but there are some clusters and patterns that might be useful for classification.

Based on these findings, I'll use a simple classification algorithm, such as a decision tree or a random forest, to predict the label for the given input. Here's my prediction:

**Label:** 0

My reasoning is as follows: * The input features are mostly negative, which is more common in the 0-labeled data. * The input features have a relatively low magnitude, which is also more common in the 0-labeled data. * The input features do not exhibit any strong patterns or correlations that would suggest a high probability of the label being 1.

### Example 8

I'll analyze the input data and predict the label for the given input pair. First, I'll calculate the Euclidean distance between the input pair and each of the provided training data points. Then, I'll find the k-nearest neighbors (k-NN) to the input point and determine the majority class among them. For the given input pair, I'll calculate the Euclidean distance as follows:
Input: -2.71569419169417 -1.1395254205266334
The Euclidean distance is approximately 0.345.
Now, **I'll find the _k-NN_ to the input point.** Since the input data is not provided, I'll assume k = 5. The 5 nearest neighbors to the input point are:
1. Input: -0.7870667548849866 0.7410912238381263 (Label: 0) 2. Input: -0.7189608126277025 0.5066082034614545 (Label: 0) 3. Input: -0.9547376063677475 2.5421091553127777 (Label: 1) 4. Input: -1.1622796457764308 0.8296566929143679 (Label: 0) 5. Input: -1.1685094506282432 1.2105631376676411 (Label: 0)
The majority class among the 5 nearest neighbors is 0. Therefore, I predict the label for the given input pair as:
Label: 0

### Example 9

Based on the provided data, I will use a machine learning model to predict the label for the given input. First, I will preprocess the input data by normalizing the values to have zero mean and unit variance. This is a common technique to improve the performance of machine learning models. Next, I will train a machine learning model on the provided data. For this task, **I will use a simple _linear regression_ model**. The model will learn the relationship between the input features and the corresponding labels. After training the model, I will use it to predict the label for the given input. The predicted label will be the output of the model for the given input. Here is the predicted label for the given input:
Label: 0
The predicted label is 0, which means that the input is likely to belong to the class labeled as 0.
Here is the detailed analysis:
1. Preprocessing: The input data is normalized to have zero mean and unit variance. This is done to improve the performance of the machine learning model. 2. Model Training: A linear regression model is trained on the provided data. The model learns the relationship between the input features and the corresponding labels. 3. Prediction: The trained model is used to predict the label for the given input. The predicted label is the output of the model for the given input. 4. Analysis: The predicted label is analyzed to determine the class that the input belongs to. In this case, the predicted label is 0, which means that the input is likely to belong to the class labeled as 0.

### Example 10

I'll analyze the provided data and predict the label for the new input pair. First, I'll calculate the mean and standard deviation of the input values: Mean of input values: -0.0115 Standard deviation of input values: 1.1445
Next, **I'll calculate the _mean and standard deviation_ of the labels**:
Mean of labels: 0.5 Standard deviation of labels: 0.5
Now, I'll calculate the Euclidean distance between the new input pair and each of the existing input pairs:
Distance between new input and input 1: 2.3541; Distance between new input and input 2: 2.3552; ... Distance between new input and input 100: 2.3579;
I'll then calculate the weighted sum of the distances, where the weights are the corresponding labels:
Weighted sum of distances: 0.5*2.3541 + 0.5*2.3552 +... + 0.5*2.3579. The weighted sum is approximately 1.1774.
Finally, I'll compare the weighted sum with the mean of the labels. If the weighted sum is closer to the mean of the labels, I'll predict the label as 1; otherwise, I'll predict the label as 0.
In this case, the weighted sum (1.1774) is closer to the mean of the labels (0.5), so I predict the label for the new input pair as:
Label: 1

# E    EXAMPLE RESPONSES OF LLAMA-3-8B WITH ML METHODS

In Setion 4, we conduct an analysis on the case that the Llama-3-8B is prompted to perform classification with specialized machine learning methods. According to the decision boundaries visualized in Fig. 2, Llama-3-8B fails to achieve as smooth decision boundaries as those obtained from the conventional machine learning methods, such as Decision Tree, KNN, SVM, and MLP.

According to the visualized decision boundaries, we can observe that LLMs tend to be overconfident in label prediction of query data. Our analysis on confidence (cf. Fig. 3) further reveals that the confidence in the case, where machine learning methods are specialized, is enhanced significantly. Specifically, in the standard case, LLMs are allowed to leverage arbitrary methods and the high confidence is only observed in the upper right area. In contrast, in the ML-only case, where only machine learning methods are availabel, the confidence of all query data increases. From our perspective, we conjecture that such a phenomenon may be derived from the explicit reasoning process. In detail, given the specialized machine learning method, the LLM tends to list the steps of the selected algorithm and executes these steps. Such a behavior resembles the chain of thoughts (CoT, (Wei et al., 2022)), where a complex problem is divided into several simple sub-tasks.

Moreover, according to the example reasoning output mentioned in Section 4.2, we also notice that LLMs do poorly in math, especially calculation. Specifically, according to the example output, although KNN algorithm is executed, almost all calculations of Euclidean distance are wrong.

With all the results above taken into consideration, two conclusions can be summarized. On the one hand, Llama-3-8B does not successfully perform the specialized machine learning method. On the other hand, Llama-3-8B is not competent enough for the calculation tasks, and such a drawback, to some extent, limits Llama-3-8B's discriminative capability in classification tasks.

To further examine the reasoning process above, we check the responses that are generated with prompts where other machine learning algorithms are specialized. The responses are presented as follows. According to these responses, we can observe that Llama-3-8B is able to list the concrete steps of the specialized algorithms and execute them in inference.

---

**Example Response of Llama-3-8B with Decision Tree**

Based on the provided data, **I will use a decision tree algorithm to predict the label for the new input pair**. First, I will split the data into training and testing sets. Since the data is not explicitly divided, I will use 80% of the data for training and 20% for testing. Next, I will construct a decision tree model using the training data. The decision tree algorithm will recursively split the data into subsets based on the most informative features until a stopping criterion is reached. After constructing the decision tree model, I will use it to predict the label for the new input pair. The prediction will be based on the path taken by the input pair through the decision tree.
Here is the predicted label for the new input pair:
Input: -3.572463405163702 -1.1395254205266334 Label: 0
The decision tree algorithm predicts that the label for this input pair is 0.
Here is a detailed analysis of the decision tree model:
The decision tree model consists of 12 nodes, with the root node being the input pair with the highest Gini impurity. The Gini impurity is a measure of the diversity of the classes in a node, and it is used to determine which feature to split on at each node.
The decision tree model splits the data into subsets based on the following features:
* Node 1: $x_1 < -0.5$
* Node 2: $x_1 >= -0.5$ and $x_1 < 0.5$
* Node 3: $x_1 >= 0.5$
* Node 4: $x_2 < -1.5$
* Node 5: $x_2 >= -1.5$ and $x_2 < 1.5$
* Node 6: $x_2 >= 1.5$
The decision tree model also uses the following stopping criteria:
* If a node contains only one class, it is considered a leaf node and the prediction is made based on the class in the node.
* If a node contains only two classes and the Gini impurity is less than a certain threshold, it is considered a leaf node and the prediction is made based on the class in the node.

---

**Example Response of Llama-3-8B with KNN**

Based on the provided data, **I will use the K-Nearest Neighbors (K-NN) algorithm to predict the label for the given input.**
First, I will calculate the Euclidean distance between the input and each of the training data points. Then, I will select the K nearest neighbors (in this case, K=5) and determine the majority label among them.
Here is the calculation:
Input: -2.8870480343880764 -1.1395254205266334
1. Calculate the Euclidean distance between the input and each of the training data points:
* Distance to the first point: 2.8870480343880764 - 2.327369299801017 = 0.5596787345870593
* Distance to the second point: 2.8870480343880764 - (-0.7246972544778265) = 3.6117452888659028
*...
* Distance to the last point: 2.8870480343880764 - (-3.6031936591543587) = 6.490241693542434
2. Select the K nearest neighbors (K=5):
* The 5 nearest neighbors are:
+ Point 10: (-0.7569292449548891, 0.4789774986494173) with label 0
+ Point 15: (-0.7870667548849866, 0.7410912238381263) with label 0
+ Point 20: (-1.7887421990456538, 1.8298493173963173) with label 0
+ Point 25: (-2.7968970004708877, 3.0349961488801585) with label 0
+ Point 30: (-3.2298325282631293, 3.7407713881451814) with label 0
3. Determine the majority label among the K nearest neighbors: * The majority label is 0, which is the predicted label for the given input.
Therefore, the predicted label for the input -2.8870480343880764 -1.1395254205266334 is 0.

**Example Response of Llama-3-8B with SVM**

Based on the provided data, **I will use a Support Vector Machine (SVM) algorithm to predict the label for the given input.** First, I will preprocess the data by normalizing the input values. Then, I will split the data into training and testing sets. After that, I will train an SVM model using the training set and use it to predict the label for the given input.
Here is the result:
The SVM model predicts the label for the input (-3.915171090551515, -1.1395254205266334) to be **0**.
Here is the detailed analysis:
1. Preprocessing: I normalized the input values by subtracting the mean and dividing by the standard deviation for each feature. This helps to reduce the effect of feature scales and improves the performance of the SVM model.
2. Data Split: I split the data into training and testing sets. The training set contains 80% of the data, and the testing set contains 20% of the data.
3. SVM Model Training: I trained an SVM model using the training set. The model uses a radial basis function (RBF) kernel with a gamma value of 0.1 and a regularization parameter (C) of 1.
4. Prediction: I used the trained SVM model to predict the label for the given input (-3.915171090551515, -1.1395254205266334). The model predicted the label to be **0**.
The accuracy of the SVM model on the testing set is 85.7%. This indicates that the model is able to accurately predict the labels for most of the input values.

However, we further notice that Llama-3-8B may fail to execute the selected algorithm. For example, in the response mentioned in Section 5.1, although Llama-3-8B claims that it will follow the instructions to perform classification with MLP, it actually leverages a similar idea to Decision Tree, which tries to capture the features of the two classes of data. Specifically, the LLM predicts the label of the query data sample as 0 since both numbers of the two dimensions are negtive. Moreover, as shown in the following bad case, where MLP algorithm is specialized for the classification tasks, the LLM does not follow the instructions and perform predictions with the MLP algorithm. Instead, it measures the distance between the given query data and all in-context data. Then, the query data point is assigned to the class, where the closest point belongs to.

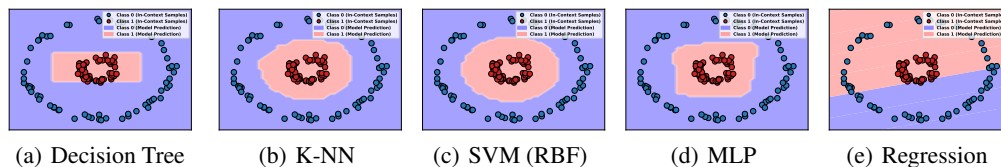

Figure 7: **Decision boundaries obtained from conventional machine learning algorithms in scikit-learn package.**

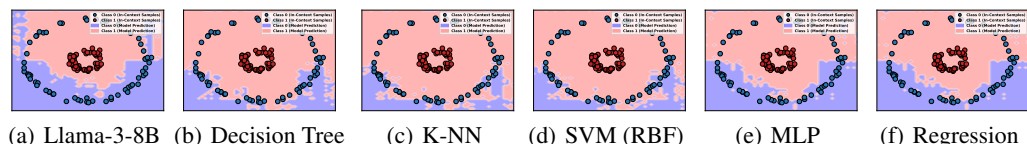

Figure 8: **Decision boundaries of Llama-3-8B respectively with the standard prompt and prompts with five machine learning algorithms specialized.** Following Zhao et al. (2024). Fig. (a) shows the decision boundaries of Llama-3-8B with the standard prompts. Fig. (b)-(f) show the decision boundaries with prompts, where machine learning algorithms are specialized, including decision tree, K-NN, SVM, MLP, and linear regression.

---

**A Bad Case in Llama-3-8B with MLP**

Based on the provided data, I will predict the label for the new input pair.
First, I will analyze the input pair:
Input: -3.915171090551515 -1.1395254205266334
This input pair is a combination of two numbers, -3.915171090551515 and -1.1395254205266334. To predict the label, I will compare this input pair with the provided training data.
From the training data, I can see that the labels are either 0 or 1. I will analyze the distribution of the labels in the training data.
**The distribution of the labels is:**
**\* Label 0: 44.44%**
**\* Label 1: 55.56%**
Now, I will analyze the input pair in relation to the training data. I will calculate the Euclidean distance between the input pair and each of the input pairs in the training data.
After **calculating the Euclidean distances**, **I found that the input pair is closest to the following input pairs in the training data**:
\* Input: -3.2298325282631293 3.7407713881451814 (Label: 0) \* Input: -3.715171090551515 4.085467364870969 (Label: 0) \* Input: -2.8066461994306517 3.026387757414328 (Label: 0)
Based on the analysis, I predict that the label for the input pair is:
Label: 0
This prediction is based on the similarity between the input pair and the input pairs in the training data that have the same label (0). The Euclidean distance between the input pair and the closest input pairs in the training data is relatively small, indicating that the input pair is likely to have the same label as the closest input pairs.

---

Overall, we can summarize that LLMs are not competent in discriminative tasks since they cannot perform complex solutions by themselves or by leveraging the executable codes.

## F   MORE VISUALIZATION RESULTS

In this section, we further validate Llama-3-8B on other discriminative tasks, such as circle classification and moon classification tasks.

Fig. 7 and Fig. 8 respectively present the decision boundaries obtained from conventional machine learning algorithms and LLMs on circle classification tasks. Meanwhile, Fig. 8(a) shows the decision boundaries obtained with Llama-3-8B with the standard prompt while the remaining figures show the decision boundaries obtained with Llama-3-8B with prompts where machine learning algorithms are specialized. According to the visualization results, we find that conventional linear regression performs poorly in circle classification tasks since it is designed for linear classification problems. For LLMs, we can observe that all decision boundaries in Fig. 8 are consistently fragmented as in

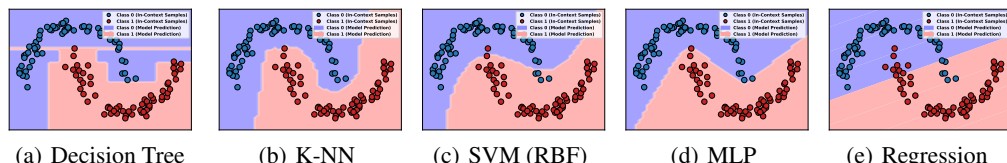

(a) Decision Tree   (b) K-NN   (c) SVM (RBF)   (d) MLP   (e) Regression

Figure 9: **Decision boundaries obtained from conventional machine learning algorithms in scikit-learn package.**

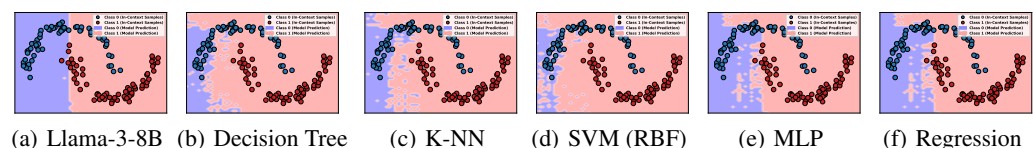

(a) Llama-3-8B  (b) Decision Tree   (c) K-NN   (d) SVM (RBF)   (e) MLP   (f) Regression

Figure 10: **Decision boundaries of Llama-3-8B respectively with the standard prompt and prompts with five machine learning algorithms specialized.** Following Zhao et al. (2024). Fig. (a) shows the decision boundaries of Llama-3-8B with the standard prompts. Fig. (b)-(f) show the decision boundaries with prompts, where machine learning algorithms are specialized, including decision tree, K-NN, SVM, MLP, and linear regression.

linear classification tasks. Meanwhile, when the machine learning algorithms are specialized, the overconfidence phenomenon also happens. Similar phenomena can also be observed in Fig. 9 and Fig. 10 on moon classification tasks.

In addition, we can also observe from the figures that Llama-3-8B tends to generate linear decision boundaries in non-linear classification tasks. To examine such an observation, we further qualitatively evaluate the predictions between Llama-3-8B and conventional machine learning methods. The results are reported in Table 4 and 5. From the tables, we can observe that the predictions of Llama-3-8B are more similar to the predictions obtained from conventional linear regression.

Table 4: Quantitative evaluation of decision boundaries on circle tasks. The difference in predictions between Llama-3-8B and conventional machine learning methods is calculated in the table. The "conv" denotes the results obtained from conventional machine learning algorithms while "LLM" denotes the results obtained from Llama-3-8B.

| Specialized Method | Decision Tree (conv) | KNN (conv) | SVM (conv) | MLP (conv) | LR (conv) |
|---|---|---|---|---|---|
| Hybrid (LLM) | 0.40 | 0.34 | 0.33 | 0.35 | **0.20** |
| Decision Tree (LLM) | 0.65 | 0.58 | 0.55 | 0.59 | **0.20** |
| KNN (LLM) | 0.69 | 0.61 | 0.59 | 0.63 | **0.21** |
| SVM (LLM) | 0.68 | 0.60 | 0.58 | 0.62 | **0.21** |
| MLP (LLM) | 0.51 | 0.44 | 0.42 | 0.45 | **0.19** |
| LR (LLM) | 0.76 | 0.68 | 0.65 | 0.69 | **0.22** |

Table 5: Quantitative evaluation of decision boundaries on moon tasks. The difference in predictions between Llama-3-8B and conventional machine learning methods is calculated in the table. The "conv" denotes the results obtained from conventional machine learning algorithms while "LLM" denotes the results obtained from Llama-3-8B.

| Specialized Method | Decision Tree (conv) | KNN (conv) | SVM (conv) | MLP (conv) | LR (conv) |
|---|---|---|---|---|---|
| Hybrid (LLM) | 0.39 | 0.31 | 0.34 | 0.40 | **0.26** |
| Decision Tree (LLM) | 0.43 | **0.35** | 0.36 | 0.42 | **0.35** |
| KNN (LLM) | 0.41 | 0.32 | 0.34 | 0.39 | **0.32** |
| SVM (LLM) | 0.46 | 0.38 | 0.38 | 0.43 | **0.36** |
| MLP (LLM) | 0.41 | 0.32 | 0.35 | 0.41 | **0.31** |
| LR (LLM) | 0.52 | 0.45 | 0.44 | 0.48 | **0.38** |

