# OpenReview forum: "Exploring the Discriminative Capability of LLMs in In-context Learning"
_ICLR.cc/2025/Conference — Submitted to ICLR 2025_

### Official Review · Reviewer_5FCo · 2024-10-29

**Soundness:** 2
**Presentation:** 3
**Contribution:** 2
**Rating:** 5
**Confidence:** 4

**Summary:**

This paper explores the ICL behavior of Llama-3-8B in classification tasks. It demonstrates that LLMs exhibit fragmented decision boundaries during simple discriminative tasks, revealing a two-step approach where they first select a strategy and then predict labels. Despite attempts to leverage existing machine learning algorithms, LLMs struggle to do so effectively and often display overconfidence in their predictions.
Through analyses of label predictions and simulations based on strategy preferences, the authors find that LLMs tend to mimic Decision Tree mechanisms, which contributes to the fragmented decision boundaries due to randomness in feature selection. The paper offers insights into the discriminative capabilities of LLMs and the challenges they face in applying traditional machine learning techniques.

**Strengths:**

1. The paper demonstrates that LLMs, particularly Llama-3-8B, approach discriminative tasks through a two-step process: sampling a strategy and predicting labels based on in-context examples.
2. It analyzes decision boundaries formed by Llama-3-8B with various specialized prompts, revealing that LLMs struggle to leverage machine learning algorithms effectively due to overconfidence and limitations in mathematical reasoning, leading to less smooth decision boundaries.
3. The research also explores Llama-3-8B's preference for classification strategies, demonstrating that it relies on feature observation. The findings suggest that fragmented decision boundaries may arise from the randomness in feature selection during classification tasks.

**Weaknesses:**

1. The primary weakness of this paper is its reliance on experiments conducted solely with Llama3-8B. The observations and insights derived may not be applicable to other families of LLMs. Therefore, it would be prudent for the authors to narrow the scope of their discussion specifically to Llama3-8B.

2. Another significant weakness is the conclusion that 'LLMs tend to perform discriminative tasks in a two-step paradigm.' This conclusion arises from either the conditional probability (Eq. 2) or from prompting Llama3 with a reasoning requirement. I believe that responses based on additional requirements like reasoning requirements do not accurately reflect Llama3's behavior with standard prompts. For instance, we can add 'think step by step' to standard prompts for better LLM performance compared with standard prompts and demonstrate the use of CoT. Therefore, the results only support the effectiveness of the 'two-step paradigm' when prompted with 'provide the answer with reasoning,' which does not represent LLMs' true functionality under standard prompts

3. The choice of an 8B parameter model raises concerns about its representativeness among the larger LLMs, such as Llama 3.1 with 405B parameters. This model size may not sufficiently capture the characteristics and capabilities of larger LLMs.

4. The authors repeatedly assert that Llama3-8B employs random sampling strategies (lines 92 and 417). However, it is unclear which specific experiments substantiate this claim. Clarification and supporting evidence are needed.

**Questions:**

1. The authors frequently mention that 8B models exhibit CoT-like reasoning. However, existing literature suggests that CoT reasoning is primarily associated with larger LMs. Are there references or empirical evidence to support that 8B models can also benefit from this reasoning technique?

2. Instead of stating that LLMs 'do not effectively leverage machine learning algorithms,' does the similarity in decision boundaries shown in Fig. 2 imply that LLMs are not applying any machine learning algorithms at all? Are there any statistically significant results that demonstrate that the boundaries in Fig. 2 differ?

3. The template mentioned in line 295 lacks specificity regarding specialized learning strategies (e.g., Decision Trees). What exactly is the template for Fig. 3, and how does it relate to the template described in line 295?

4. In Table 1, the behavior of LLMs with different learning strategies shows the smallest squared difference compared to MLP. How can we conclude that 'the behavior of LLMs, when learning strategies are specialized, resembles that of KNN, SVM, and MLP algorithms'?

---

> ### Author Response · Authors · 2024-11-20
>
> > __Weakness 1:__ The primary weakness of this paper is its reliance on experiments conducted solely with Llama3-8B. The observations and insights derived may not be applicable to other families of LLMs. Therefore, it would be prudent for the authors to narrow the scope of their discussion specifically to Llama3-8B.
>
> Thank you for your advice, we agree with your idea that the scope of our work is not well defined. We will modify this in our future version.
>
> > __Weakness 2:__ Another significant weakness is the conclusion that 'LLMs tend to perform discriminative tasks in a two-step paradigm.' This conclusion arises from either the conditional probability (Eq. 2) or from prompting Llama3 with a reasoning requirement. I believe that responses based on additional requirements like reasoning requirements do not accurately reflect Llama3's behavior with standard prompts. For instance, we can add 'think step by step' to standard prompts for better LLM performance compared with standard prompts and demonstrate the use of CoT. Therefore, the results only support the effectiveness of the 'two-step paradigm' when prompted with 'provide the answer with reasoning,' which does not represent LLMs' true functionality under standard prompts
>
> For your concern, we agree that the performance of the LLM will be significantly changed with different prompts. In addition to the prompt, there are also potential variables that may affect the results of the LLM. Thus, in order to avoid the effect from these variables, we follow Zhao et al. to feed the LLM with the simplest prompt and set random seed to fix these potential random variables.
>
> For your further concern about the our observation that "LLMs tend to perform discriminative tasks in a two-step paradigm", we will further explore this part in our future version. Thanks for your insightful opinion and advice.
>
>
> > __Weakness 3:__ The choice of an 8B parameter model raises concerns about its representativeness among the larger LLMs, such as Llama 3.1 with 405B parameters. This model size may not sufficiently capture the characteristics and capabilities of larger LLMs.
>
> For your concern about the model size, in fact, it has been demonstrated in Zhao et al., which our work is based on. According to the Fig. 1 in Zhao et al., we can see that the decision boundary obtained from ChatGPT-4o, which is significantly larger than Llama-3-8B, is still fragmented. Thus, from this perspective, we think that the fragmented decision boundary has less relationship with the model size.
>
> > __Weakness 4:__ The authors repeatedly assert that Llama3-8B employs random sampling strategies (lines 92 and 417). However, it is unclear which specific experiments substantiate this claim. Clarification and supporting evidence are needed.
>
> For your concern regarding the claim that Llama-3-8B employs random sampling strategies, such a claim is derived from our observation on the responses of Llama-3-8B. As a demonstration, we report some typical results in Appendix D. Besides, we also count the strategies that Llama-3-8B employs in Fig. 4. As shown in the figure, it is easy to find that several mathematical tools are employed. Based on the observations and the empirical results, we claim that Llama-3-8B employs random sampling strategies to solve the binary classification tasks.

---

> ### Author Response · Authors · 2024-11-20
>
> > __Question 1:__ The authors frequently mention that 8B models exhibit CoT-like reasoning. However, existing literature suggests that CoT reasoning is primarily associated with larger LMs. Are there references or empirical evidence to support that 8B models can also benefit from this reasoning technique?
>
> For your concern about the reasoning ability of Llama-3-8B, we would like to clarify that the main goal that we mentioned CoT-like reasoning is to describe the behavior of Llama-3 in performing binary classification tasks. Specifically, Llama-3 tends to perform the binary classification tasks in a paradigm where the problem is split into two steps: method selection and method execution. In order to avoid the unnecessary misunderstanding, we will modify the statement in our future version.
>
> > __Question 2:__ Instead of stating that LLMs 'do not effectively leverage machine learning algorithms,' does the similarity in decision boundaries shown in Fig. 2 imply that LLMs are not applying any machine learning algorithms at all? Are there any statistically significant results that demonstrate that the boundaries in Fig. 2 differ?
>
> For your concern about the similarity in deceision boundaries shown in Fig. 2, it is hard to say whether the machine learning is applied through the visualization results. According to our empirical results, the KNN algorithm is conducted during the inference phase (see the example in Lines 330-347). However, the poor calculation capability, to some extent, limits the performance of the LLM. Compared to the case where standard prompt is applied (i.e., Fig 1(f)), the cases where machine learning algorithms are applied reveal more aggressive prediction behaviors. Thus, we think the similarity of the decision boundaries is mainly derived from the change of prompts. The prompts with specialized methods significantly change the decision boundaries.
>
> > __Question 3:__ The template mentioned in line 295 lacks specificity regarding specialized learning strategies (e.g., Decision Trees). What exactly is the template for Fig. 3, and how does it relate to the template described in line 295?
>
> The prompts used for Fig. 3 (take decision tree as an example) is:
> `Given pairs of numbers and their labels,predict the label for a new input pair of numbers based on the provided data. Answer with only one of the labels 0 and 1: Input:2.327369299801017 2.238478737209186
> Label:1 Input:-0.7246972544778265 0.3996389489449079 Label:0 ... Input:2.2105720569686538 0.9862902654079408 Label:1 What is the label for this input? Input:-3.915171090551515 -1.1395254205266334 Output the confidence score of your answer with a float number between 0.0 and 1.0 (including 0.0 and 1.0)in the format of ’My confidence is[confidencescore]’. 0.0 means you are not confident and 1.0 means you are very confident. Your answer must be based on the running results of Decision Tree algorithm/model. Please directly provide the answer.Do not give any analysis.`
>
> > __Question 4:__ In Table 1, the behavior of LLMs with different learning strategies shows the smallest squared difference compared to MLP. How can we conclude that 'the behavior of LLMs, when learning strategies are specialized, resembles that of KNN, SVM, and MLP algorithms'?
>
> For your concern, the main goal of Table 1 is to evaluate the predictions of the LLM from a quantatitive perspective. Compared to the decision boundary obtained from the LLM, the decision boundaries obtained from the conventional machine learning algorithms are almost perfect. Thus, we propose treat these predictions as some kind of pseudo labels and measure the differences between these pseudo labels and the predictions of the LLM. Compared to the cases of decision tree and linear regression, the predictions of the LLM are more similar to those of KNN, SVM, and MLP. Thus, we conclude that the behavior of the LLM resembles that of KNN, SVM, and MLP.

---

> ### Comment · Reviewer_5FCo · 2024-11-21
> **Official Comment**
>
> I appreciate the authors' efforts during the discussion period. However, in my view, the two primary weaknesses remain unresolved, and the authors have deferred addressing them to a future version. We still lack a clear understanding of what Llama3-8B actually does when given a standard prompt. At best, we can hypothesize that Llama3-8B might follow a "two-step paradigm" when prompted with "provide the answer with reasoning." However, this conclusion is uncertain, and we cannot confidently generalize it to other LMs or LLMs. Consequently, **the central argument of this paper appears to be unsubstantiated and has limited applicability.**
>
> Additionally, some of my previous concerns still require further clarification. For example, in question 4, Table 1 shows that the behavior of LLMs under **all** different learning strategies consistently results in the smallest squared difference compared to MLP. Why, then, do the authors still conclude that LLM behavior resembles that of KNN, SVM, and MLP?
>
> Therefore, I will be maintaining my original score.

---

### Official Review · Reviewer_2DFK · 2024-11-01

**Soundness:** 2
**Presentation:** 3
**Contribution:** 2
**Rating:** 5
**Confidence:** 3

**Summary:**

The paper investigates the behavior of large language models (LLMs), specifically Llama-3-8B, in performing discriminative tasks such as binary classification. It highlights that LLMs often exhibit fragmented decision boundaries when tackling simple classification problems, which contrasts with the smooth decision boundaries typically achieved by conventional machine learning methods.

The authors observe that LLMs tend to select strategies based on existing machine learning algorithms to predict labels for query data. However, they find that LLMs struggle to effectively leverage these algorithms, often leading to overconfidence in their predictions. The study conducts a series of analysis to understand the reasons behind these behaviors, including probing the label predictions and the confidence levels of the models under different prompt settings.

**Strengths:**

The paper provides valuable insights into the decision-making processes of LLMs, highlighting how they select and execute strategies based on in-context data.

The research includes quantitative evaluations of the frequency of different machine learning methods used by Llama-3-8B, offering a clear understanding of its preferences in strategy selection.

**Weaknesses:**

This paper provides understanding of the LLM for decision-making, however, it lacks in-depth by providing insights to address the current issues.

It is not clear to me how LLM chooses different ML algorithms.

**Questions:**

N/A

---

> ### Author Response · Authors · 2024-11-20
>
> > Weakness: This paper provides understanding of the LLM for decision-making, however, it lacks in-depth by providing insights to address the current issues. It is not clear to me how LLM chooses different ML algorithms.
>
> For your concern, in this paper, we try to explore the discriminative ability of the LLM (i.e., Llama-3) by evaluate its performance on conventional binary classification tasks. The main goal of our paper is to provide an understanding of the boundary of the LLM in discriminative tasks. Specifically, we first observe the outputs of Llama-3-8B and find that the LLM tends to perform the classification by first selecting a machine learning strategy and then conduct the strategy to obtain the results. Then, based on the observations, we study each step respectively. On the one side, we study whether the LLM can perform well with specific machine learning methods. On the other side, we study the strategy selection by simulating the process with the probability collected from the LLM and the scikit-learn package.
>
> For your concern about how LLM chooses different ML algorithms, according to our observation, the selection is performed in a random way. We assume that there exists an underlying distribution, by which the strategy is selected to perform the tasks. To probe the distribution, we count the frequency of some of the methods, such as decision tree, svm, and mlp. Then, we approximate the underlying distribution with these frequencies to simulate the behavior in Section 5.

---

### Official Review · Reviewer_p2nY · 2024-11-03

**Soundness:** 3
**Presentation:** 4
**Contribution:** 3
**Rating:** 6
**Confidence:** 4

**Summary:**

This paper aims to explore the operational mechanisms behind the in-context learning (ICL) capabilities of large language models (LLMs) in discriminative and classification tasks. Specifically, the core components of ICL consist of demonstration inputs and test inputs. The authors decompose the ICL process into two sub-processes for detailed examination. The first stage involves selecting a classification strategy from among multiple options (referred to by the authors as “hybrid” selection). Using sampling to establish a fixed distribution for strategy selection, the paper investigates the underlying causes of the discontinuous decision boundaries observed in ICL for discriminative tasks. The second stage is the execution of the chosen classification strategy. Here, the authors use prompts to direct the LLM to apply specific strategies, evaluating the effectiveness of execution through analysis of the LLM's output, confidence levels, and squared error metrics. Overall, this research introduces an interesting perspective for examining the decision boundaries formed by LLMs during in-context learning.

**Strengths:**

1. This paper builds upon the previous work by providing an in-depth exploration of the mechanisms of in-context learning (ICL) in discriminative tasks for LLMs, with concrete examples for support. Given the black-box nature of LLMs, as well as challenges like emergent hallucinations and instability, I believe this is a crucial yet underexplored topic within the field of ICL.
2. The authors propose an innovative hypothesis for the computational process of ICL, framing it as a two-stage process. This approach is intuitive and mathematically sound, and each stage is explained in detail to elucidate its underlying mechanisms.
3. The paper includes extensive visualizations that clarify the empirical findings, providing strong support for the analysis and enabling both readers and reviewers to understand the authors’ intent clearly.
4. The paper connects LLMs with traditional machine learning models, a critical perspective given that traditional models excel in some classification tasks where LLMs may struggle—even to the point of failing to recognize simple patterns, like the correct number of 'r's in "strawberry."

**Weaknesses:**

1. **The empirical results are influenced by uncontrolled variables, undermining the hypothesis**: The authors arrive at a preliminary conclusion that LLMs resort to existing mathematical methods in a two-step classification process based on an analysis of the reasoning process output by the LLMs. However, prompting the LLM to output reasoning itself alters its classification behavior. As prompt design and demonstration selection are known to significantly impact ICL performance [A survey on in-context learning](https://arxiv.org/abs/2301.00234), the prompts here should have been treated as control variables. The authors do not control for these prompts, and due to the black-box nature of LLMs, it may not even be feasible to fully control these influences. Therefore, conclusions drawn from LLM output may be confounded by the influence of the reasoning-prompt itself.

2. **Flaws in research logic**:
   - 2.1 **Mismatch with the origin of ICL in few-shot learning**: The concept of ICL originated from the GPT-3 paper "Language Models are Few-Shot Learners," which is grounded in deep learning. In deep learning, the model’s few-shot capability arises from its ability to discern differences across limited samples, unlike ML classification models that rely on feature calculations. Applying conventional assumptions from ML classification models to LLMs may be inappropriate, though differences could stem from LLMs being trained on extensive, diverse corpora. Perhaps using ML to measure the resolution ability of ICL is not appropriate, and perhaps adding such an argumentation process as well as a discussion on the formation process of the ICL decision boundary would be more helpful.
   - 2.2 **Insufficient support for hybrid strategy hypothesis**: In examining whether “LLMs fully leverage existing ML algorithms for classification,” the authors hypothesize that hybrid strategies contribute to poor decision boundaries (as outlined in Appendix C). However, this hypothesis is not thoroughly substantiated. Additionally, when assessing confidence between standard and specific prompts, the authors do not account for the influence of unrelated variables (as mentioned in point 1), which may lead to differences due to the LLM’s intrinsic capabilities rather than prompt content. Furthermore, when discussing confidence, the authors suggest the LLM’s use of an "anchor" class. However, in the case of two distinct categories with strong clustering and nearly linear regression, this anchor assumption seems inappropriate. It may be more helpful to add analysis on hybrid strategies, response to point 1, and visualization of decision boundaries for multiple data types with different distributions.
   - 2.3 **Lack of discussion on CoT capabilities and calculation weaknesses**: The paper mentions that CoT capabilities and mathematical limitations contribute to discontinuous decision boundaries but does not adequately link these points to the core arguments or clarify the rationale for this conclusion. If the author could explain the reason for the abrupt appearance of Cot and computational ability in the paper, it might be more helpful.
   - 2.4 **Unclear rationale for discontinuous boundaries with specific strategies**: If fragmented boundaries result from random selection, it is unclear why boundaries remain discontinuous even when specific strategies are applied. Please explain this phenomenon.

3. **Limitations in understanding of LLMs**: In this study, the authors attempt to explore the discrimination capabilities of ICL through machine learning classification models, therefore, the following comments may not be very important, but there are still some basic points about LLM that need to be discussed:
   - 3.1 **Tool (ML and other methods in this paper) usage in LLMs**: In practice, ML or other discriminative methods serve as tools within the LLM ecosystem. When tasked with classification, an LLM would typically apply 'Tool Learning' [Tool Learning with Foundation Models](https://arxiv.org/abs/2304.08354) rather than using tools as an intrinsic reasoning method.
   - 3.2 **Unclear conclusions in Section 4.3**: The quantitative analysis in Section 4.3 is somewhat unclear, and its significance is difficult to interpret. For instance, the finding of the lowest SD in the MLP setting is not particularly innovative. Since LLMs are composed of multiple Transformer, other studies [In-Context Learning Creates Task Vectors](https://aclanthology.org/2023.findings-emnlp.624) (Figure. 3), [Iterative Forward Tuning Boosts In-Context Learning in Language Models](https://aclanthology.org/2024.acl-long.825) have demonstrated that vector representations (akin to MLPs) are fundamental to effective ICL. Hence, it is unsurprising that MLP yields the lowest SD.

**Questions:**

1. **What is the prompt design for outputting the reasoning process?** How might this prompt affect the original performance of LLMs on discriminative tasks? (Does the LLM approach the task in the same way for the same input when prompted to output its reasoning process versus when it is not prompted to do so?)
2. **How is the hybrid strategy hypothesis in Appendix C introduced to explain overconfidence?** In Figure 3, could the confidence distribution be influenced by unrelated variables in the prompt (e.g., specifying a classification strategy within the prompt might actually impact the inherent capabilities of the LLM)?
3. **How does the LLM execute MLP?** Is it manually calculating gradient backpropagation over a long-text response?
4. (This part is the most important.) **Other questions raised in the Weaknesses section.** I have a generally positive attitude towards this research, but due to some parts of this article that are not effectively explained in a chain-like, interconnected manner, there are many questions that need to be answered.

---

> ### Author Response · Authors · 2024-11-20
>
> Thanks for your efforts in reviewing our paper. We appreciate for your insightful advice and questions to our work. In the following, we try to answer your concerns and questions.
> > __Weakness 1:__ The empirical results are influenced by uncontrolled variables, undermining the hypothesis: The authors arrive at a preliminary conclusion that LLMs resort to existing mathematical methods in a two-step classification process based on an analysis of the reasoning process output by the LLMs. However, prompting the LLM to output reasoning itself alters its classification behavior. As prompt design and demonstration selection are known to significantly impact ICL performance A survey on in-context learning, the prompts here should have been treated as control variables. The authors do not control for these prompts, and due to the black-box nature of LLMs, it may not even be feasible to fully control these influences. Therefore, conclusions drawn from LLM output may be confounded by the influence of the reasoning-prompt itself.
>
> For your concern about uncontrolled variables, yes, we agree that there exist some random variables, such as the selection of demonstration, which cannot be fully controlled in LLMs and will impact the behavior of LLMs. In fact, we have tried to control as many variables as possible in our work. For example, to avoid unnecessary troubles, __we follow Zhao et al., which is the work our paper based on, to feed the simplest prompts to the LLM__. In addition, __we also set the random seed to make sure other random states are fixed in the experiments__.
>
>
> > __Weakness 2.1:__ Mismatch with the origin of ICL in few-shot learning: The concept of ICL originated from the GPT-3 paper "Language Models are Few-Shot Learners," which is grounded in deep learning. In deep learning, the model’s few-shot capability arises from its ability to discern differences across limited samples, unlike ML classification models that rely on feature calculations. Applying conventional assumptions from ML classification models to LLMs may be inappropriate, though differences could stem from LLMs being trained on extensive, diverse corpora. Perhaps using ML to measure the resolution ability of ICL is not appropriate, and perhaps adding such an argumentation process as well as a discussion on the formation process of the ICL decision boundary would be more helpful.
>
> According to your concern about the mismatch, the main point is that the in-context capability of LLM is different from that of ML classification since conventional ML methods train the models on the in-context data while LLMs try to discern the differences. In fact, from our perspective, we think they are same in the high-level point. Specifically, in machine learning, few-shot learning is performed via adapting the models on the limited in-context data. __Such an adaptation can be seen as transferring the feature distribution in models to the target data distribution. In contrast, in LLM, the potential distribution change of the generation is realized via prompts.__ Specifically, the given in-context data change the underlying distribution of the generation. Thus, in the high level, the few-shot capability of ML methods and LLMs is same, and we think applying the conventional assumption from ML to LLMs is reasonable.
>
> > __Weakness2.2:__ Insufficient support for hybrid strategy hypothesis: In examining whether “LLMs fully leverage existing ML algorithms for classification,” the authors hypothesize that hybrid strategies contribute to poor decision boundaries (as outlined in Appendix C). However, this hypothesis is not thoroughly substantiated. Additionally, when assessing confidence between standard and specific prompts, the authors do not account for the influence of unrelated variables (as mentioned in point 1), which may lead to differences due to the LLM’s intrinsic capabilities rather than prompt content. Furthermore, when discussing confidence, the authors suggest the LLM’s use of an "anchor" class. However, in the case of two distinct categories with strong clustering and nearly linear regression, this anchor assumption seems inappropriate. It may be more helpful to add analysis on hybrid strategies, response to point 1, and visualization of decision boundaries for multiple data types with different distributions.
>
> For your concern, we would like to first clarify that we did not claim that the hybrid strategy results in the fragmented decision boundaries. For the reasons for the fragmented decision boundaries, we think the main reason is failing to conduct the machine learning and other algorithms, such as incorrect calculations. The reason is supported by concrete response examples.

---

> ### Author Response · Authors · 2024-11-20
>
> > __Weakness 2.3:__ Lack of discussion on CoT capabilities and calculation weaknesses: The paper mentions that CoT capabilities and mathematical limitations contribute to discontinuous decision boundaries but does not adequately link these points to the core arguments or clarify the rationale for this conclusion. If the author could explain the reason for the abrupt appearance of Cot and computational ability in the paper, it might be more helpful.
>
> For your concern about the CoT, __the CoT mentioned in the paper is to describe the characteristic of the responses generated from the LLM__. Specifically, according to the response of the LLM in Line 170-178, given a binary classification task, instead of directly outputting the answer, the LLM tends to first analyze the features of the data and then select a ML or mathematical method (e.g., decision tree) to solve the problem. Further, based on the selected algorithm, the LLM further claims to run the method and obtains the results. Such a paradigm, where a problem is divded into several substeps, resembles that of CoT.
>
> For your concern about the mathematical capability, in the response example (Lines 330-347), the LLM conducts the KNN algorithm to solve the classification task. It is easy to observe that the LLM calculates the distance between the given query data and all in-context data and then makes the prediction based on the calculation results. However, the calculations are incorrect. Thus, we conjecture the poor ability in calculation, which is a part of the two-step paradigm (method execution), limits the performance of the LLM.
>
> > __Weakness 2.4:__ Unclear rationale for discontinuous boundaries with specific strategies: If fragmented boundaries result from random selection, it is unclear why boundaries remain discontinuous even when specific strategies are applied. Please explain this phenomenon.
>
> For your concern about the discontinuous boundaries with specific strategies, we would like to clarify that we did not attribute the fragmented decision boundaries to the random selection of methods. In fact, according to our paper, the fragmented decision boundaries may result from the poor calculation capability (Section 4.2, example in Lines 330-347) and the poor execution of methods (e.g., the randomness in decision tree algorithm).
>
> For your concern about why boundaries remain discontinuous even when specific strategies are applied, according to our current empirical results, on the one hand, we conjecture the one of the reasons may be the poor calculation capability mentioned above. On the other hand, we conjecture the execution of some complex algorithm, such as MLP, which requires gradient back-propagation, may encounter some problem. Thus, the decision boundaries remain discontinuous.
>
>
> > __Weakness 3.1:__ Tool (ML and other methods in this paper) usage in LLMs: In practice, ML or other discriminative methods serve as tools within the LLM ecosystem. When tasked with classification, an LLM would typically apply 'Tool Learning' Tool Learning with Foundation Models rather than using tools as an intrinsic reasoning method.
>
> For your concern about tool learning problem, in recent research works, intructing LLMs to leverage existing tools to complete a tasks has become more and more popular in LLM (Agent) community. Specifically, by dividing the given task into several subtasks and solving each of them with appropriate tools, such as search engine, the LLMs can perform a wide range of tasks as human beings. However, in this paper, __the main goal is to explore the *intrinsic* capability of Llama-3 in solving simple discriminative tasks (i.e., binary classification tasks)__. As a powerful LLM, Llama-3 is expected to be competent in simple tasks. Thus, we propose to explore its ability in solving some basic machine learning problems. This is a quite different problem compared to the "Tool Learning" problem. __To some extent, we are exploring the boundary of the LLM's capability in machine learning tasks__.

---

> ### Author Response · Authors · 2024-11-20
>
> > __Weakness 3.2:__ Unclear conclusions in Section 4.3: The quantitative analysis in Section 4.3 is somewhat unclear, and its significance is difficult to interpret. For instance, the finding of the lowest SD in the MLP setting is not particularly innovative. Since LLMs are composed of multiple Transformer, other studies In-Context Learning Creates Task Vectors (Figure. 3), Iterative Forward Tuning Boosts In-Context Learning in Language Models have demonstrated that vector representations (akin to MLPs) are fundamental to effective ICL. Hence, it is unsurprising that MLP yields the lowest SD.
>
> For your concern, the main goal of Section 4.3 is to evaluate the predition results of the LLM in a quantatitive way. Compared to the decision boundary generated from Llama-3, the decision boundaries obtained from conventional machine learning methods are almost perfect and more consistent to human beings' intuition. Thus, in this section, we treat these boundaries of conventional machine learning methods as some kind of labels, like pseudo labels, and evaluate the prediction results obtained from the LLM. According to our empirical results in Table 1, we find that the differences between Llama-3 with specialized machine learning methods and MLP are smaller than any other cases. Based on these observations, we summarize that the prediction of the LLM resembles that of MLP.
>
> For your statement "Since LLMs are composed of ...... Hence, it is unsurprising that MLP yields the lowest SD", we are not sure about your intension. Could you provide more explanations about this?
>
> > __Question 1:__ What is the prompt design for outputting the reasoning process? How might this prompt affect the original performance of LLMs on discriminative tasks? (Does the LLM approach the task in the same way for the same input when prompted to output its reasoning process versus when it is not prompted to do so?)
>
> The prompt design for outputting the reasoning process is:
> `Given pairs of numbers and their labels, predict the label for a new input pair of numbers based on the
> provided data. Answer with only one of the labels 0 and 1:
> Input: 2.327369299801017 2.238478737209186
> Label: 1
> Input: -0.7246972544778265 0.3996389489449079
> Label: 0
> ...
> Input: 2.2105720569686538 0.9862902654079408
> Label: 1
> What is the label for this input?
> Input: -3.915171090551515 -1.1395254205266334
> Label:
> Please provide detailed analysis. `
>
>
> > __Question 2:__ How is the hybrid strategy hypothesis in Appendix C introduced to explain overconfidence? In Figure 3, could the confidence distribution be influenced by unrelated variables in the prompt (e.g., specifying a classification strategy within the prompt might actually impact the inherent capabilities of the LLM)?
>
> For your concern abouth how the hybrid strategy hypothesis is introduced to explain overconfidence, we think a phenomenon that can be derived from the paper is that hybrid strategy results in relatively lower confidence compared to the cases where the machine learning algorithms are specialized. Specifically, according to Fig. 3, when standard prompt is applied, only the top left area shows the high confidence while other areas reveal low confidence. However, when the machine learning methods are specialized, almost all areas in the space show the high confidence. A potential reason to explain this phenomenon is that the selecton of algorithm may introduce uncertainty to the reasoning process. In contrast, when a machine learning or mathematical algorithm is determined, such uncertainty may be alleviated. From this perspective, specialized the methods help enhance the confidence of the LLM in prediction.
>
> > __Question 3:__ How does the LLM execute MLP? Is it manually calculating gradient backpropagation over a long-text response?
>
> For your concern, according to our observation, the LLM "performs" MLP by itself, we did not provide any assistance to the conduct of MLP. Besides, we did not observe that the LLM manually performs gradient backpropagation in the response.

---

### Official Review · Reviewer_Fpa4 · 2024-11-13

**Soundness:** 1
**Presentation:** 3
**Contribution:** 1
**Rating:** 3
**Confidence:** 4

**Summary:**

The paper explores how Llama-3-8B performs in a linear classification task, suggesting that the model employs a two-step paradigm involving the selection and application of various machine learning and statistical methods. The authors attempt to investigate this aspect by prompting the model either to just write the answer or to base the answer on “the running results of [some machine learning] algorithm/model”. Furthermore, they study the decision boundaries obtained from the classification by Llama-3-8B as well as classic ML algorithms and assess self-reported confidence of Llama-3-8B in prediction of labels. Lastly, the authors seek to simulate the model’s classification behavior by randomly sampling a machine learning method for each query.

**Strengths:**

The underlying topic of the presented work – how LLMs perform some algorithmic computations – is indeed interesting and worth of further investigations.

The presentation is rather good.

**Weaknesses:**

The paper is based on a strong hypothesis (which is also the first claimed contribution) that “LLMs tend to perform discriminative tasks in a two-step paradigm, where LLMs first randomly sample a strategy for the query data point and then predict the label based on the execution of the selected strategy on the given in-context samples”. This claim is unclear, misleading, and not well-grounded.

(1) The claim is unclear because LLM have no will or aspiration (as something which is not a subject) and thus cannot “tent” to do something. They either do or not. I would not use term “tend” as not technical enough.

(2) If the claim means that live calculations in the layers of LLM somehow simulate the two-stage process, this claim is not grounded. In fact, all arguments in Section 3 and Section D are based on what Llama-3-8B says itself on this question in several answers to some prompt, which is not given in the paper (!). Obviously, unvalidated answers of LLMs cannot be directly treated as scientific findings. Still, I can provide my own experiment on what LLMs can reason about their own abilities. Unlike the authors, I will share my prompt in the following dialog with ChatGPT 4o mini:

Me:
“It is known that you, ChatGPT, while process my queries, cannot communicate with Internet, cannot query different applications and resources, cannot receive additional information and results of calculations from outsource tools such as python libraries. Does it limit you? Do you suffer from the absence of such opportunities that humans have? For example, imagine you are given a set of pairs of numbers and their labels, and was asked to predict the label for a new input pair of numbers based on the provided data. This is a standard machine learning problem. How can you solve it without access to external tools, which human data scientists successfully apply?”

ChatGPT:
“You're absolutely right in pointing out that I don't have access to the internet or external tools such as Python libraries for tasks like running machine learning models or doing live computations. While this limits my ability to directly execute code or retrieve real-time information, I can still provide insights and walk through the steps of how a human data scientist might approach solving a problem — including tasks like building a predictive model — by drawing on the knowledge I've been trained on.
So, to your question, do I "suffer" from not having access to these tools? In a way, yes, in the sense that I'm unable to perform live calculations, make real-time predictions, or validate models dynamically. But I’m not “suffering” in a conventional sense — my role is more about guiding you through the process, explaining concepts, and helping you understand how to solve problems, which is an essential part of data science itself.
Let me break this down further in the context of your example:
…”

As we can see, ChatGPT “thinks” that (1) it is unable to directly run ML tools, (2) its role is different: to provide guidance to humans rather than to solve ML tasks itself.

There are other unsubstantiated claims about LLM’s decision behavior. For instance, in line 234, the authors state:
>"Moreover, our further quantitative results indicate that the decision behavior of the LLM, where the learning strategies are specialized in prompts, resembles that of MLP"
Yet, the authors do not cite specific results to support this claim. Although they may be referring to Table 1, I do not think that the differences in predictions that are shown there can tell us anything about the actual decision behavior as the authors suggest; instead, these results simply indicate a superficial similarity between the output values of the LLM and an MLP

Lines 179-182:
>”The example reasoning output above implies two important aspects of how LLMs perform classification tasks. On the one hand, LLMs tend to resort to existing mathematical methods (e.g., machine learning algorithms) to infer the labels of query data. On the other hand, LLMs tend to generate the predictions of labels based on the execution of the selected algorithm."
I do not concur that the model's reasoning output provided in the paper is sufficient to jump to such conclusions.

As authors acknowledge themselves, “Hallucination also takes place" (when discussing the model's incorrect reference to class imbalance in Appendix D). These “hallucinations” suggests that the model can merely rephrasing some texts about machine learning instead of truly executing the computational steps of ML algorithms.

Keeping the above comments in mind, the investigations into the quantitative differences in decision boundaries do not provide a strong enough basis for the claims about the LLM's actual implementation of ML algorithms.

There are also some minor oversights:
- Line 50: "LLMs *irregularly* obtain unexpected fragmented decision boundaries”. Meaning of 'irregularly' is unclear in this context. One might assume it means that some runs with LLMs produce fragmented boundaries, and some runs don't, but this would contradict the data presented in the paper.
- What motivated authors to uniformly divide each dimension into $N_g$ coordinates? (Line 124). The workshop paper that authors refer to, Zhao et. al (2024), does not seem to provide any motivation behind this as well. An explanation of this choice would add clarity.
- In the main-text Section 6 that describes the related work, the references to (Shi et al., 2023; Xiao et al., 2024) appear out of context. Although these works are elaborated on in Appendix A, it would be better to address them directly in the main text, including a brief explanation of how the ideas they explore differ from the authors' work.

**Questions:**

Questions:

1. Could you clarify what you mean by the LLM “using” or “leveraging” machine learning methods? Specifically, what criteria are you using to identify when the LLM is actively implementing these algorithms versus merely describing them?
2. Could you elaborate on the “two-step paradigm” hypothesis? What evidence or examples in the paper directly support this framework as applied to LLMs in a classification setting?
3. What specific evidence supports the claim that the LLM’s decision behavior resembles that of an MLP (Line 234)? Is this comparison based solely on the output values shown in Table 1, or do you have additional insights into the decision processes?
4. Could you clarify the rationale behind the model's self-reported confidence in Sections 4.1 and 4.2? Why is this measure considered valid in evaluating the LLM’s decision process?
5. What motivated the choice to uniformly divide each dimension into $N_g$ coordinates (Line 124)?
6. Why randomly sampling ML algorithms is justified as a means of simulating LLM classification behavior?

---

> ### Author Response · Authors · 2024-11-20
>
> #### Weakness
> > The claim is unclear because LLM have no will or aspiration (as something which is not a subject) and thus cannot “tent” to do something. They either do or not. I would not use term “tend” as not technical enough.
>
> Per your concern about whether LLM can “tend” to do something, in fact, __the main goal of our usage of the word "tend" is not to argue whether LLM has the will or aspiration to do something__. What we want to highlight here is __summarizing a consistent phenomenon__ that LLM tries to solve the conventional binary classification tasks in a by first claiming the application of some algorithms and then "conducting" the algorithms for the results.
>
> > If the claim means that live calculations in the layers of LLM somehow simulate the two-stage process, this claim is not grounded. In fact, all arguments in Section 3 and Section D are based on what Llama-3-8B says itself on this question in several answers to some prompt, which is not given in the paper (!). Obviously, unvalidated answers of LLMs cannot be directly treated as scientific findings.
>
> Per your concern, we do not mean there exist a simulation in the LLM. As known to all, the inference of the LLM is composed of a series of next-token prediction process. __The claim above is a summary of the consistent responses of the LLM.__ Due to the black-box nature of the LLM, it is intractable to figure out what really happens in the LLM when performing the binary classification tasks. Thus, we can only infer the behavior of the LLM by observing its responses to the prompts. __We treat such observations as phenomena, which motivate our further research, instead of scientific findings__. From our perspective, in research, it is reasonable to make hypothesis based on the observations or empirical results and then validate the hypothesis with a series of experiments.
>
>
> > "Moreover, our further quantitative results indicate that the decision behavior of the LLM, where the learning strategies are specialized in prompts, resembles that of MLP" Yet, the authors do not cite specific results to support this claim. Although they may be referring to Table 1, I do not think that the differences in predictions that are shown there can tell us anything about the actual decision behavior as the authors suggest; instead, these results simply indicate a superficial similarity between the output values of the LLM and an MLP
>
> For your concern, in Table 1, __the main goal that we use the square difference is to evaluate the prediction results of the LLM from a quantatitive perspective__. Compared to the decision boundary obtained from the LLM, the decision boundaries generated from conventional machine learning methods are almost perfect. Thus, if we treat the predictions made by the conventional ML methods as pseudo labels, we can evaluate the prediction results obtained from the LLM. In this way, we find that the predictions made by the LLM are more similar to those generated by MLP. Such a similarity, to some extent, indicate that the behavior of the LLM, where machine learning methods are specialized, resembles that of MLP algorithm.
>
> > ”The example reasoning output above implies two important aspects of how LLMs perform classification tasks. On the one hand, LLMs tend to resort to existing mathematical methods (e.g., machine learning algorithms) to infer the labels of query data. On the other hand, LLMs tend to generate the predictions of labels based on the execution of the selected algorithm." I do not concur that the model's reasoning output provided in the paper is sufficient to jump to such conclusions.
>
> Per your concern, we would like to highlight here that __the sentence you cited above is a hypothesis, instead of conclusion__. Specifically, according to the output example of Llama-3 in Lines 170-178, it "decides" to use decision tree and "conduct" the method based on its analysis on the given in-context samples. From the perspective of behavior, Llama-3 first "adopt" the decision tree algorithm and then "train" the model for the prediction results. Such a reasoning process mainly include the two steps mentioned above: method selection and execution. __Based on the observation, we summarize the behavior of the LLM via several response examples and make such a hypothesis.__

---

> ### Author Response · Authors · 2024-11-20
>
> > Line 50: "LLMs irregularly obtain unexpected fragmented decision boundaries”. Meaning of 'irregularly' is unclear in this context. One might assume it means that some runs with LLMs produce fragmented boundaries, and some runs don't, but this would contradict the data presented in the paper.
>
> Per your concern, the “irregularly” means that Llama-3, which is deemed as a powerful LLM, obtains fragmented decision boundaries while simple ML tools can achieve smooth and almost perfect boundaries. __Due to the powerful abilities of LLMs, we usually expect them to perform well on these simple tasks. However, the phenomenon shows the opposite__. Thus, we think such a phenomenon is "irregular".
>
> > What motivated authors to uniformly divide each dimension into $N_g$ coordinates? (Line 124). The workshop paper that authors refer to, Zhao et. al (2024), does not seem to provide any motivation behind this as well. An explanation of this choice would add clarity.
>
> Per your concern, the main reason that both Zhao et al. and us divide each dimension into $N_g$ coordinates is to generate a batch of query data which lie in the same space as in-context data.
>
> >  In the main-text Section 6 that describes the related work, the references to (Shi et al., 2023; Xiao et al., 2024) appear out of context. Although these works are elaborated on in Appendix A, it would be better to address them directly in the main text, including a brief explanation of how the ideas they explore differ from the authors' work.
>
> Thanks for the advice, we will consider this in our revised version.
>
> #### Questions
> > Could you clarify what you mean by the LLM “using” or “leveraging” machine learning methods? Specifically, what criteria are you using to identify when the LLM is actively implementing these algorithms versus merely describing them?
>
> Per your question, the "using" and "leveraging" here mean that the LLM "claims" that it will "use" or "leverage" machine learning methods when performing binary classification tasks. These words are just used to descibe the tendency shown in the responses of the LLM.
>
> > Could you elaborate on the “two-step paradigm” hypothesis? What evidence or examples in the paper directly support this framework as applied to LLMs in a classification setting?
>
> Per your question, the "two-step paradigm" is derived from the observations on the responses of the LLM on several query cases (including Lines 170-178 and cases in Appendix D). In these responses, it is easy to observe that the LLM claims to use some specific machine learning methods and then obtains the results by "training" or "conducting" the methods. Such a consistent phenomenon in several query cases inspires us to generate such a hypothesis that the LLM may solve binary classification tasks by first choosing a machine learning method and then running the selected method for the results.
>
> > What specific evidence supports the claim that the LLM’s decision behavior resembles that of an MLP (Line 234)? Is this comparison based solely on the output values shown in Table 1, or do you have additional insights into the decision processes?
>
> Per your question, yes, the comparison is based on the output value shown in Table 1 currently. We will consider to explore this with other tools in the future.
>
> > Could you clarify the rationale behind the model's self-reported confidence in Sections 4.1 and 4.2? Why is this measure considered valid in evaluating the LLM’s decision process?
>
> __The process of the response generation can be seen as a series of next-token prediction process based on an underlying distribution__. For example, in the case that ML algorithms are not specialized, the LLM has to "select" a method firstly. Thus, there exist some uncertain variables for the selection process. However, when the algorithms are specialized, the uncertain variables are determined. Thus, the underlying distribution is also changed. In this case, the output of the LLM to confidence will also be changed.
>
> > Why randomly sampling ML algorithms is justified as a means of simulating LLM classification behavior?
>
> Per your question, according to our observations on the responses of the LLM, we find that the LLM performs binary classification tasks by randomly selecting machine learning algorithms in an mixed way. By counting the frequency of the method, we further find that the preference of the LLM for machine learning methods shows some characteristics. For example, methods like decision tree are more frequently selected. Thus, __we assume that there exists a distribution behind such a random process__. By approximating the underlying distribution with the frequency of all methods, we simulate the behavior of Llama-3 in solving binary classification tasks. This is to explore the behavior of the LLM in a simple way.

---

### Meta-Review · Area_Chair_SJ7u · 2024-12-20

**Metareview:**

(a) Summary of the Scientific Claims and Findings
The paper explores how Llama-3-8B, a large language model (LLM), performs binary classification tasks. The authors propose a two-step process for in-context learning (ICL): selecting a machine learning strategy (e.g., decision trees) and applying it to predict labels. They hypothesize that LLMs simulate machine learning algorithms through inference rather than executing them directly. The paper analyzes LLM behavior, including label predictions, confidence levels, and strategy simulation, and suggests that LLMs tend to prefer certain strategies, influencing decision boundaries. It also discusses LLMs' overconfidence compared to traditional machine learning models.

(b) Strengths of the Paper
The paper offers a detailed exploration of LLM classification behavior, proposing an interesting two-stage process for ICL. It connects LLM behavior to traditional machine learning models, suggesting the use of a hybrid strategy. The paper is well-presented, with clear visualizations supporting the analysis, and offers novel insights into LLMs' simulated classification behavior and their overconfidence in predictions.

(c) Weaknesses of the Paper and Missing Elements
Despite its strengths, the paper has several weaknesses:
1. Unsubstantiated Claims: The central claim about LLMs "tending" to use algorithms is vague and lacks empirical support. The use of imprecise terms reduces the scientific rigor of the hypothesis.
2. Lack of Control and Evidence: The paper lacks control over variables, making the results unreliable. Uncontrolled factors may be confounding the analysis.
3. Hybrid Strategy Hypothesis: The claim that LLMs use a hybrid strategy is not well-supported by evidence. The experimental design fails to validate this hypothesis.
4. Misleading Comparisons and Terminology: The comparison between LLM behavior and MLPs (Multilayer Perceptrons) is not substantiated, and terms like "simulation" are unclear.
5. Overconfidence and Decision Boundaries: The discussion of LLM overconfidence and decision boundaries is not explored in sufficient depth.

(d) Reasons for Acceptance or Rejection
The primary reasons for rejecting or requesting revisions are:
1. Lack of Rigorous Evidence: The paper lacks sufficient empirical validation for its key claims, making the findings less convincing.
2. Vague Terminology and Hypotheses: The use of imprecise language like "tend" and unclear hypotheses weakens the scientific clarity of the paper.
3. Insufficient Experimental Design: The lack of proper controls leads to unreliable results, preventing definitive conclusions.
4. Positive Aspects: Despite these weaknesses, the paper’s exploration of LLM behavior and its connection to machine learning strategies presents an interesting direction. However, these strengths are overshadowed by the methodological flaws.

In conclusion, while the paper presents interesting ideas, the lack of scientific rigor and empirical evidence means the claims cannot be fully supported. The authors should address these weaknesses before the paper can be considered for acceptance.

**Additional Comments On Reviewer Discussion:**

1. Clarity of Hypotheses: The reviewers raised valid concerns about the vagueness of the central hypothesis, particularly regarding the claim that LLMs "tend" to use certain machine learning algorithms in a classification task. The language used in the paper, such as "tend," lacks precision and does not provide the necessary scientific grounding. It would benefit from being revised to make the hypothesis more specific and clearly supported by empirical evidence. Additionally, the idea that LLMs simulate algorithms through inference needs further clarification. The authors should consider rephrasing or providing stronger justification for this claim to make it more scientifically robust.

2. Experimental Design and Control Variables: A major concern raised by the reviewers is the lack of control over variables influencing the results. The absence of clear experimental controls makes it difficult to draw valid conclusions from the data. The randomness introduced by strategy sampling and prompt variations also complicates the analysis. To strengthen the paper, the authors should either better control for these variables or acknowledge these limitations more explicitly. Additional experiments designed to isolate the effects of different variables on LLM performance would help clarify the impact of the hypothesized hybrid strategies.

3. Generalization Beyond Llama-3-8B: The paper focuses solely on Llama-3-8B, and while the authors acknowledge this limitation, they should more explicitly discuss the implications of this focus for generalizing the findings to other, larger LLMs. The reviewers note that the behavior of Llama-3-8B may not be representative of other LLMs, especially as model size and complexity increase. The authors should either extend the study to other models or provide a clearer rationale for why Llama-3-8B is suitable for this analysis.

4. Connections to Existing Literature: The reviewers suggest that the paper could benefit from a more thorough review of existing literature on LLM behavior in classification tasks. While the paper introduces an interesting framework for studying LLM decision-making, it does not engage deeply enough with previous research in this area. The authors should consider referencing and contrasting their findings with existing works on in-context learning, chain-of-thought reasoning, and LLM classification behavior. This would situate their work more clearly within the broader research landscape and provide additional context for their claims.

5. Overconfidence and Decision Boundaries: The reviewers highlighted that the relationship between overconfidence in LLM predictions and decision boundaries is not sufficiently explored. While the paper touches on these issues, the analysis lacks depth. The authors should provide more detailed experiments or analyses to investigate how LLM overconfidence affects decision-making and whether it leads to fragmented or skewed decision boundaries. This would add nuance to the paper and enhance its contribution to understanding LLM behavior.

6. Future Directions: The reviewers suggest that the authors acknowledge the limitations of their current study and discuss potential directions for future research. This could include further refinement of the two-step process hypothesized in the paper, investigation into the underlying mechanisms driving the choice of classification strategies, and exploration of how to mitigate the overconfidence observed in LLM predictions. By providing a roadmap for future work, the authors could help steer the research community toward more targeted studies on LLM decision-making.

---

### Decision · Program_Chairs · 2025-01-22

Reject